# Design, Manufacturing, and Testing of a New Concept for a Morphing Leading Edge using a Subsonic Blow Down Wind Tunnel

**DOI:** 10.3390/biomimetics4040076

**Published:** 2019-12-02

**Authors:** David Communier, Franck Le Besnerais, Ruxandra Mihaela Botez, Tony Wong

**Affiliations:** Laboratory of Applied Research in Active Controls, Avionics and AeroServoElasticity LARCASE, ÉTS, Montréal, QC H3C 1K3, Canada; david.communier.1@ens.etsmtl.ca (D.C.); franck.le-besnerais@estaca.eu (F.L.B.); tony.wong@etsmtl.ca (T.W.)

**Keywords:** morphing leading edge, wind tunnel testing, morphing wing

## Abstract

This paper presents the design and wind tunnel test results of a wing including a morphing leading edge for a medium unmanned aerial vehicle with a maximum wingspan of 5 m. The design of the morphing leading edge system is part of research on the design of a morphing camber system. The concept presented here has the advantage of being simple to manufacture (wooden construction) and light for the structure of the wing (compliance mechanism). The morphing leading edge prototype demonstrates the possibility of modifying the stall angle of the wing. In addition, the modification of the stall angle is performed without affecting the slope of the lift coefficient. This prototype is designed to validate the functionality of the deformation method applied to the leading edge of the wing. The mechanism can be further optimized in terms of shape and material to obtain a greater deformation of the leading edge, and, thus, to have a higher impact on the increase of the stall angle than the first prototype of the morphing leading edge presented in this paper.

## 1. Introduction

This paper presents the morphing of the leading edge (LE) of a wing with the main goal of developing a morphing camber system and integrating it within the wing of a UAS-S4 Ehécatl. The development of a morphing trailing edge (MTE) system has been presented in [1]. The effectiveness of this MTE system has been demonstrated experimentally in the Price–Païdoussis wind tunnel by comparing it to the effectiveness of a rigid aileron [2]. It was concluded that the MTE could replace an aileron, as it was able to improve the efficiency of the wing. However, in order to improve the lift on drag (L/D) ratio of the wing over the entire wing, the MTE alone was not enough, as seen in Figure 1, where results are expressed in terms of drag coefficient (C_D_) variation with lift coefficient (C_L_). From these results, it was able to be observed that for all lift coefficient values, the drag coefficient measured was found to be higher for the MTE than for a fixed wing. For this reason, it was necessary to combine the MTE with a morphing leading edge (MLE) in order to obtain a morphing camber system.

In Figure 2, NACA0012 airfoil and NACA4412 airfoil shapes are superimposed. The only difference between these shapes is the camber, while the thickness remains the same along their chord. One airfoil shape can change to another shape via morphing of the trailing edge (TE) and of the LE, while a constant central section is maintained. In Figure 3, a gain in the drag coefficient was found for a lift coefficient greater than 0.367 when the shape of the wing changed from an NACA0012 to an NACA4412 airfoil for an angle of attack greater than 2.97°.

We found in [2] that for an NACA0012 airfoil, the displacement of a rigid aileron will have the effect of moving the C_L_ variation with the angle of attack curve to the left or to the right according to the direction (up or down). In fact, if a down displacement of the LE is performed for a given angle of attack, the lift increases. We also found in [1] that a rigid aileron and an MTE induce a decrease in the stall angle. The use of an MLE allows for the delaying of the stall angle of the wing, thus compensating for the weakness of the MTE.

Several mechanisms of MLEs have already been studied in the literature. For example, Sodja [3] and Rudenko [4] have carried out bench tests of MLE systems.

The Sodja mechanism allowed the LE to be morphed but it required a very high force from its linear actuator. This study aimed to validate the idea that an MLE system can reach the desired shape for drag and noise reduction.

Rudenko′s study presented another MLE concept. In his study, the numerical results expressed obtained by finite element analysis (FEA) were compared with the experimental results in terms of the MLE deformations. The MLE mechanism used a circular actuator coupled with internal articulations.

Radestock [5] and Takahashi [6] have developed mechanisms for MLEs whose performances were tested in a wind tunnel.

Radestock indicated that lift does not change during MLE deformation. This study did not have any effect on the stall angle, since the largest angle of attack studied was 8.68°.

Takahashi proposed a mechanism combining the MLE and the MTE. This combined mechanism was tested in a wind tunnel, and the results indicated that the deformation of the camber could increase the lift coefficient by 1 compared to the lift coefficient values before deformation. However, bumps were observed on the lower surface of the test wing which could have increased its drag. Takahashi confirmed that the combination of the MTE with the MLE would allow the lift coefficient to significantly increase. A study of the drag of the system is needed to determine its effectiveness compared to a conventional wing.

Our Laboratory of Applied Research in Active Controls, Avionics, and AeroServoElasticity (LARCASE) team has a long history in the field of morphing wings. Thus, the LARCASE team collaborates with major players in the aeronautical engineering sector of Montreal. Through the Consortium of Research and Innovation in Aerospace in Quebec (CRIAQ) projects, several studies have been conducted with major research consortiums (CRIAQ 7.1 [7,8] and CRIAQ MDO 505 [9,10,11,12,13,14]) and internal projects have been conducted at the LARCASE laboratory, such as one project on the morphing of a wing equipped with an ATR-42 airfoil [15].

## 2. Design of the MLE

The MLE had to be a simple and easy-to-manufacture system. We therefore limited ourselves to the use of materials and actuators common in large model unmanned aerial vehicle (UAV) designs because of the fact that this system is mainly intended for UAV usage. The wing airfoil used for the design of this system was the NACA0012. This airfoil was symmetrical and thus allowed us to identify if a variation in its aerodynamics performances according to the angle of attack resulted from a manufacturing defect (dissymmetry) or from the behavior of the MLE system. An airfoil thickness of 12% of the chord allowed for servomotor installation inside the wing.

To facilitate the deformation of the wing surface of the MLE system, the wing surface does not contribute to the structural strength of the wing. Hence, the main spar of the wing had to be designed to withstand all the aerodynamic loads that the wing would be subjected to during tests in the wind tunnel. Figure 4 shows the internal structure of the wing, including its main spar.

The MLE deformation is given by the deformation of three ribs, as seen in Figure 4, with the control rod linking these ribs. A servomotor-type actuator moves the tip of the LE via a control arm, while the flexibility of the rib is ensured by six slits regularly spaced between the main spar and the anchor point at the LE. The widths of the slits were calculated according to the desired deformation while the remaining thickness of the rib should have been able to withstand the shear forces induced by the aerodynamic pressures around the wing. Figure 5 represents the central rib of the wing in which the servomotor is fixed.

The servomotor moves a rod vertically. This rod is connected to the ribs that are desired to deform. Thus, all three ribs connected by the rod move together thanks to the control arm, as seen in Figure 6.

The MLE only has one control arm, as the rigid surface of the wing ensures that there is no twist in the MLE and that the three ribs move together with the same displacement.

To allow the control arm to move the LE through the main spar, the main spar is composed of two beams. One beam is connected to the inner surface of the wing and the other beam is connected to the upper surface of the wing. As the structure is designed to have symmetrical behavior in the wind tunnel, the two beams have the same dimensions (Figure 7).

In order to allow the LE to morph, slits were placed on the surface of the wing which corresponded to those on the ribs (Figure 8).

However, the presence of slits degrades the flow of air around the wing. To solve this problem, the wing was covered with a sheet of heat-shrinkable plastic such as MonoKote^®^. The sheet was placed in such a way that the motion of the LE was left free, in order to allow the morphing of the LE. This sheet contributed to the formation of slight humps at the level of the slits. This design, which involved covered slits, was not perfect, but as seen in Figure 9, it generated less drag than when leaving the slits opened and just a little more drag than when leaving the slits closed (i.e., when the slits did not have humps on their surface). The values presented in Figure 9 were obtained during wind tunnel tests on the MTE at a speed of 15 m/s [1]. It is important to mention the fact that the configuration which kept the slits closed cannot be used because it does not allow the LE to be morphed.

In order to test a prototype in the wind tunnel, a wing with the integrated MLE system was designed and manufactured. This wing was provided with a circular base which allowed for its fixation in the test chamber of the wind tunnel. Figure 10 shows this wing prototype equipped with an MLE system.

## 3. Structural Analysis of the MLE System

The size of the slits is usually determined by four parameters: the width of the slit *l*, the depth of the slit *p*, the distance between the slit and the LE *L*, and the airfoil thickness *e* (Figure 11). The parameter *t* represents the thickness of material remaining at the bottom of a slit and is defined by Equation (1), as seen in Figure 11
(1)t=e−2∗p.

Since L≫t, according to the compliant mechanism’s theory [16], we can assume that the bottom of the slits acts as a pivot link. Hence, the maximum rotation allowed by the slits can be calculated from Figure 11, as follows:(2)$$MLE\;\mathrm{angle}=\tan^{-1}\left(\frac{l}{p}\right).$$

For the determination of *t*, FEA using CATIA V5 was performed on the design of the MLE. In the FEA, the maximum value of *t* is dependent on the maximum torque of the servomotor. To ensure the maximum strength of the ribs, the value of *t* was determined so that the servomotor could deform the ribs until the slits closed on themselves while maintaining the values of aerodynamic loads applied to the MLE. In order to take into account aerodynamic loads applied to the wing when using FEA with CATIA V5, a methodology was developed at LARCASE [17,18]. Figure 12 shows the MLE with the aerodynamic load applied on its surface. The aerodynamic loads were obtained with XFLR5 software and then were imported into CATIA V5.

The mesh of the pieces that deform and that of the other pieces (rib, arm, and rod) are fine in order to obtain an accurate distribution of the constraints. The mesh of each part was generated with the tool “OCTREE Tetrahedron Mesh” with parabolic elements. The rib mesh had a “global size” of 0.15 inches and a “local size” around the slits of 0.02 inches. The embedding of the assembly was placed on the two holes corresponding to the position of the main spar. The rib was connected to the control rod with the function “slider connection mesh”. The control rod had a mesh of a “global size” of 0.05 inches. The control arm was connected to the control rod with the function “pressure fitting connection mesh”. It was modelled with a mesh which had a “global size” of 0.1 inches and with a local mesh at both ends which had a “local size” of 0.05 inches. The control arm was connected with the servomotor with two functions of “slider connection mesh”. The servomotor had the largest global mesh of 0.25 inches because it did not deform. The servomotor was fixed to the rib with four function “contact connection meshes”. (Figure 13). In order to impose an angle of rotation for the servomotor head, a “rigid virtual part” was added on the control arm, connected with the servomotor head. A function “user-defined restraint” was placed on the “rigid virtual part” with a restrain on the rotation corresponding to the axis of the servomotor head. Finally, an “enforced displacement” was placed on the “user-defined restraint” with the rotation that was needed for the FEA. In Figure 13, the rotation angle was set to −5° to obtain a down motion of the leading edge. The mesh in Figure 13 had a total of 61,393 elements.

During the FEA, the input was the “angle of the servomotor head” in degrees and the output was the “constraint” in the wood parts (ribs and control arm) in MPa (color from blue to red), as shown in Figure 14. The components that were the most stressed were the “control arm” and the “ribs”. The stresses on the rib were concentrated on the slits. As wood was used for the MLE design, the maximum stresses before the rib broke were around 70 MPa. The maximum stresses found in the control arm were around 30 MPa. Hence, the control arm was able to send the rotation of the servomotor to the MLE. Concerning the slit, this FEA did not gave good results as the material in CATIA V5 was an isotropic one but the “wood” used for the ribs was “orthotropic”, and “orthotropic material 3D” was not available for computation. As the sizes of the slits were set following our design with the MTE, the flexibility of the ribs should have been good. A static test of the flexibility of the ribs before wing manufacturing was done to ensure that the MLE would deform with the full amplitude desired.

The contribution of each slit in the displacement *y* of the LE tip was able to be calculated with Equation (3), i.e.,
(3)$$y=\frac{\left(l\times L\right)}{p}$$

Table 1 shows the dimensions of the parameters *e*, *t*, *l*, *p*, *L*, *MLE* angle, and *y*, of the six slits. These dimensions correspond to a “reference” wing chord of 10 inches (25.4 cm). The definitions of these parameters are given in Figure 11.

By use of Equation (4) and the parameter values shown in Table 1, the maximum displacement of the LE was able to be calculated by adding the contribution from each of the six slits, i.e.,
(4)$$\mathrm{Maximum}\;\mathrm{MLE}\;\mathrm{displacement}\;=\;\;\sum_{i=1}^{6}y_{i}\;=\;0.322\;\mathrm{in}\;(8.18\;\mathrm{mm})$$

The value of this total displacement value was able to be numerically validated using the FEA of a morphed rib. To obtain the maximum displacement of the MLE, its corresponding angle needed to be found. Thus, the angle of the servomotor head was tuned to find the maximum displacement angle when all the slits were closed (Figure 15). Following this operation design using CATIA V5, the servomotor head angle that gave the maximum displacement of the MLE was found to be 10°.

Thus, for an angle of the servomotor head of 10°, the displacement of the MLE was obtained by calculating the *z* component of the displacement of the MLE tip node. As seen in Figure 16, the absolute maximum displacement computed using FEA is 0.333 was found to be (8.46 mm). A relative error of 3.4% was found for the MLE maximum displacement calculated with Equation (4) versus the maximum displacement obtained using CATIA V5 software. This small relative error shows that the method of calculation used for the maximum displacement of the MLE can be considered a very good method. This method allowed us to design the MLE with a desired maximum displacement without needing an FEA. This method might therefore be used in future research.

As seen above, no optimization process was used for this dimensioning, as the objective of this work until now was to obtain a functional MLE system. After the demonstration of the functionality of the MLE system, optimization to increase the amplitude of the deformation and to further to reduce the required power for its obtention was realized. The displacement of the MLE tip obtained using FEA (Figure 16) was compared with the MLE tip displacement obtained used the test wing during wind tunnel tests, as seen in the following sections.

## 4. Experimental Setup of the MLE System

All tests were carried out in the Price–Païdoussis subsonic wind tunnel of the LARCASE [19,20] (Figure 17). This tunnel is an open circuit wind tunnel 40 feet in length. The wind tunnel consists of a centrifugal fan, a diffuser and settling chamber, a contraction section, and a working section. The dimensions of these different sections are indicated in Figure 18. The test section measured 3 feet in width, 2 feet in height and 4 feet in length (Figure 19). In this wind tunnel, testing was done for speeds between 6 m/s and 35 m/s. The measurements presented in this section were made during wind tunnel testing at a speed of 20 m/s and at an air density of 1.18 kg/m³. A more detailed description of the wind tunnel and its method of calibration can be found in [21].

The aerodynamic loading scales installed in the wind tunnel measured the forces and moments experienced by the objects studied in the test section [22]. The aerodynamic loading scales consisted of a turntable controlled by a stepper motor (Table 2) assembled on a force and torque (F/T) sensor. An Omega 160 F/T sensor (Table 3) from ATI Industrial Automation was used in the aerodynamic loading scales designed and manufactured in house.

This aerodynamic loading scales allowed the dynamic reading of the loads and thus their measurements for several flight conditions in terms of angles of attack, Reynolds numbers, and Mach numbers without stopping the running of the wind tunnel between each measure. In addition, the sensor was able to read high forces and moments (Table 3), and, by use of filters, was also able to read very small forces (0.01 N). The ability to read very small forces is necessary to measure the drag forces acting on the test wings. The high force reading was used to measure the drag of large objects such as a surveillance radar from FLIR Systems, whose drag measurements were also made in the LARCASE Price–Païdoussis wind tunnel. Figure 20 shows the internal structure of the aerodynamic loading scales.

The upper part of the aerodynamic loading scales was a hollow tray used for the installation of objects to be studied in the wind tunnel (Figure 21). In order to be installed on the aerodynamic loading scales, the objects to be tested had to have their discs on which they are installed have a diameter of 10.7 inches and a base 0.5 inches thick. The disc had to be fixed at the basis of the aerodynamic loading scales with four screws.

The wing used for subsonic wind tunnel testing had a 12 inch span and a part of it (of 0.5 inches) was embedded into the base shape as a disc; thus, the span exposed to the airflow was 11.5 inches. The wing chord was chosen to be 10 inches.

The LE could not be morphed over the entire span; its MLE section had a length of 9.5 inches and covered over 20% of the wing chord. Figure 22 shows the leading edge of the MLE where the difference of shapes between the fixed and the morphing part of the leading edge can be seen at the root of the wing.

Two wingspan sections, each of them 1 inch at the wing root and at the wing tip, could not be morphed. These two sections made it possible to identify the impact of discontinuities between the morphing section and the fixed sections. This impact was around 0.002 on the value of the drag coefficient. The impact of the discontinuities could also not be neglected especially at small angles of attack, but these discontinuities are also present for classical LE flap or a slat. This impact is not a weakness of the MLE system with respect to the classical system already used.

A servomotor controls the MLE. The servomotor used is HITEC HS-5685MH, which allows for control of up to a maximum torque of 179 oz/in (12.9 kg/cm) and a maximum current of 2600 mA. The servomotor used an ‘Arduino-uno’ controller. A multi-meter was added to the controller to monitor the current consumed by the servomotor in order to avoid over-consumption that could lead to the servomotor malfunction.

A LabVIEW interface is presented in Figure 23. This interface was divided into different sections. The first section allowed the readings of the forces, their variations with respect to the disc angle, and their variation with the command sent to the servomotor (Figure 24). All the graphs showing these variations are plotted in real time in one LabVIEW section. These plotted graphs were able to be exported to a text file for analysis of the results. Another section of the LabVIEW interface was dedicated to the control of the disc angle (Figure 25). The LabVIEW interface was also linked with a PhidgetStepper Bipolar HC control board in order to control the stepper motor of the aerodynamic loading scales (Table 4).

Another section of this LabVIEW interface was dedicated to the control of the servomotor (Figure 26). Hence, the angle of the servomotor was controlled, then trimmed, and followed by a feedback on its position, as the mechanical position 0 of the servomotor was not accurate with respect to the position zero of the control surfaces. An electrical adjustment of the servomotor position (trim) was needed to get an accurate position 0 of the control surfaces.

The results presented in the following section were obtained using the angle of the MLE as the command sent with LabVIEW to the servomotor.

## 5. Wind Tunnel Test Results

The lift and drag coefficients were calculated from wind tunnel test measurements using the equations
(5)$$C_{L}=\frac{L}{Q\times S}\;,$$
(6)$$\;\;C_{D}=\frac{D}{Q\times S}\;\;,$$
where the dynamic pressure *Q* is given by
(7)$$Q=\frac{1}{2}\times \rho \times v^{2}\;,$$
and where *L* and *D* are the lift and drag forces measured by the aerodynamic loading scales and *S* is the wing surface.

The performance curves of $C_{L}$ and $C_{D}$ were able to be plotted with respect to the angle of attack to observe the influence of the MLE system on the performance of the wing.

Figure 27 shows the results expressed as the variation of the lift coefficient as a function of the angle of attack of the wing. The black curve represents this result for the wing without using the MLE system. The gray curves represent the result of deformation for the wing with the MLE system for the angles of rotation of the servomotor head which are equal to 18° and −18°. The angles of 18° and −18° correspond to the maximum MLE system deformations. As the main goal was to identify the best performances of the MLE for the stall angle, these maximum MLE deformation angles were chosen for this study. Note that during a deformation of the MLE, the stall angle of the wing varies. Thus, a positive (downward) deformation will delay the stall angle of the wing (18°) and a negative (upward) deformation will advance the wing stall angle (12°). This result obtained for the MLE’s behavior corresponds to the classical behavior of a slat. Figure 28 shows the variation of the lift coefficient as a function of the deformation of the MLE. It was found that the lift coefficient remained constant for MLE angles smaller than the stall angle of the wing, which was approximately 15°. This result obtained for the MLE system shows its improved behavior over a conventional slat behavior which slightly degrades the variation of the lift coefficient when moving the LE [23].

Figure 29 shows the result in terms of variation of the drag coefficient as a function of the angle of attack of the wing. The black curve represents this result for the wing without MLE deformation. The gray curves represent this result for the wing with MLE deformations of 18° and −18°. The maximum inaccuracy of the drag force measurement going up to 0.1 N caused an important fluctuation in the reading of the forces; drag values during tests can be seen to have varied between 0.15 N and 0.6 N for angles ranging −5° to 10°. Despite this inaccuracy, drag curve variation with the angle of attack was able to be traced and its trend can be further observed in Figure 29. Apart from the stall angle, which was different for each MLE angle configuration, the drag coefficient did not show significant variation between each MLE angle changed configuration. As seen in Figure 30, drag coefficient variation was more due to measurement imprecision than an effect of the MLE.

Since drag reading cannot be improved at this time, drag analysis cannot be pushed any further to quantify it accurately. However, it was observed that the morphing of the LE did not increase the wing drag. As the lift coefficient did not vary according to the MLE angle and the drag coefficient did not show significant variation according to the MLE angle, the MLE should not have significantly affected the L/D ratio. Due to imprecision in the measurement of drag forces which was found to be smaller than 0.4 N, the variation in the reading of drag forces close to 0 N had too much effect on the L/D ratio in order to obtain a readable curve as that shown in Figure 31.

In order to compare the experimental deformation with the deformation previously calculated, the MLE tip displacement was measured. Before adding the sheeting of the wing (ribs alone), the MLE could achieve a maximum displacement of 0.313 inches (8 mm). Once the wing was manufactured, a maximum displacement of 0.157 inches (4 mm) was obtained for both directions of the MLE deformation (positive and negative) during static experimental tests. This experimental deformation is twice as small as the calculated deformation of 0.322 inches (8.18 mm). This fact indicates that after its manufacture, the MLE was not able to reach its calculated maximum displacement.

To increase this displacement, “finishing” at the slots needed to be improved in order to obtain less obstruction against the deformation. In addition, the experimental angle of the servomotor was found to be larger (18°) than the calculated angle (10°). This observation therefore indicates a larger experimental twist of the internal control mechanism than the twist obtained using the FEA.

## 6. Aerodynamic Simulation of the Wing

The performances of the wing obtained in the wind tunnel were compared with the ANSYS Fluent simulation of the wing. ANSYS Fluent allows for the obtaining of fast and accurate computational fluid dynamics (CFD) results [24]. The performance criteria considered to compare the experimental results with the simulation results were the lift and drag coefficients, which were dependent on the angle of incidence of the airfoil with respect to the flow. The fluid used was air with a constant density of 1.225 kg/m³ and a constant viscosity of 1.7894e-05 kg/(m∙s). Then, we specified a flow velocity equal to the one used in the wind tunnel, i.e., 20 m/s. The reference values used for the wing geometrical parameters were an area of 0.0625 m² and a length of 0.254 m. To obtain comparable results, the distance of the LE tip deformation was defined as 4 mm (Figure 32), which corresponds to the maximum displacement obtained experimentally.

The modeling of the wing was carried out in three phases [25], and these phases were (1) the design of a 3D model using Catia V5 software (Figure 33), (2) the determination of a fluid domain and its mesh computation with ANSYS Meshing (Figure 34), and (3) the simulation of aerodynamic coefficients with ANSYS Fluent (Figure 35).

The fluid domain used for the simulation corresponded to the fluid in the test section of the wind tunnel. It was generated with the DesignModeler of ANSYS Fluent. The inlet was rounded in order to reduce the volume of the fluid domain and to reduce the computation time.

Morphed wing airfoil coordinates were obtained from an FEA of the system with CATIA V5 (Figure 14). From this morphed airfoil it was possible to design a simplified model of the wing which did not consider the slits, was experimentally validated in the wind tunnel, and allowed a first study approximation.

The mesh in Figure 36 was designed using the Meshing tool of ANSYS Fluent. The size function “proximity and curvature” was used in order to get a good resolution around the wing (relevance of −10). An inflation around the wing was used in order to get the finest mesh at the surface of the wing (transition ratio of 0.272, five layers, and a growth rate of 1.2). Finally, a face sizing around the sharp corners (trailing edge, slits, and bumps) was used in order to avoid convergence problems using an element size of 1 mm. For this first simulation, the mesh had around 3,652,638 elements with a satisfying quality (average orthogonality of 0.85 and average skewness of 0.23). Using this configuration, the mesh size was calculated in 10 to 15 min depending on the computer execution speed.

The model used in ANSYS Fluent was a combination of the Standard k-ε (two equations) [26,27,28,29] with standard wall functions [30,31] because this represents a good compromise between calculation time and quality of results. The two equations model was chosen due to the computing capabilities of our computers.

Once a fluid domain was designed and a suitable mesh calculated, it was possible to run simulations at different angles of incidence in order to obtain lift (Figure 37) and drag (Figure 38) variations with angle of attack of the wing thanks to the monitors in ANSYS Fluent.

The difference between the stall angle calculated with CFD simulations and the stall angle obtained experimentally during the wind tunnel tests was found to be 1°, which corresponds to a difference in maximum lift coefficient value of 0.212. The experimental drag variation with the angle of attack seems to have been shifted with respect to the numerical drag variation with the angle of attack. Indeed, given the observation of the drag variation with the angle of attack of the MLE at 0° deformation (Figure 39), there is a shift of the axis of symmetry of 3 degrees relative to the vertical axis at an angle of attack of 0°. The shifting of the drag curve was probably due to the experimental setup, but the exact cause needs to be still determined. As the airfoil used was a NACA0012, the drag variation with the angle of attack should have been symmetrical with respect to the vertical axis at an angle of attack of 0°. The drag variation with the angle of attack is plotted in Figure 40, where a vertical offset between the experimental and numerical values can be seen; this indicates that the prototype wing generated more drag than the ANSYS Fluent modeling. In order to obtain modeling which is faithful to the wind tunnel tests, the surface of the wing model needed to be degraded at the LE.

Grooves and bumps were added to the surface of the wing’s LE (Figure 41) to simulate the behavior of the uneven surface of the model due to the slits. The grooves and bumps were defined with a half-circle profile with a radius of 0.5 mm. There were five of each on the inner surface and five of each on the upper surface for a total of 10 grooves and 10 bumps. The size of the mesh elements around the grooves and the bumps was fixed at 0.25 mm. The size of the mesh elements around the rest of the wing was 1 mm.

Using this new model, which was closer to the real manufactured wing LE than the previous one, we can see in Figure 42 that the maximum lift coefficient had values obtained during the simulation which were close to those values obtained during wind the tunnel tests with an error of 0.1% for the maximum lift coefficient value, while the stall angle obtained in the simulation can be seen to have been 1° smaller (17°) than the stall angle obtained following the wind tunnel tests (18°). The modification (consisting of grooves and bumps added to the model) made it possible to obtain a numerical lift coefficient which was close to its experimental value, in the sense that a numerical stall angle was found to be close to the experimental stall angle. For drag variation with the angle of attack, by considering an offset of 3° for its experimental value, the simulated and measured drag coefficient values were found to be very close to each other (Figure 43). There was a difference for angle of attack close to 0° that seems to have been due to wind tunnel measurements. It can therefore be concluded that this new model allows a good estimate of the lift and drag coefficients of a wing with an MLE system.

Figure 44 shows the pressure around the wing in Pascal units for angles of attack of 0°, 10°, and 18° with the MLE at 18°, where the airspeed is 20 m/s and a reference p0 is of 101 325 Pa. The red color corresponds to an area with high pressures, the orange and yellow colors correspond to areas with low pressure, and the blue color corresponds to an area with high depressurization. For an angle of attack of 0°, the pressures and velocities were the same above and below the airfoil. This behavior was due to the symmetrical shape of the airfoil NACA0012. For positive angles, the pressure can be seen to have been high below the airfoil and low above the airfoil. This behavior allowed the wing to generate the lift required for the aircraft to fly. Figure 45 shows the norm of the velocity vector around the wing in m/s for angles of attack of 0°, 10°, and 18° with the MLE at 18°, where the airspeed is 20 m/s. The red vector corresponds to the highest velocity and the blue vector corresponds to the slowest velocity. In Figure 45, the acceleration of the airflow is clearly visualized on the top of the LE for angles of attack of 10° and 18°. This acceleration combined with a velocity near 0 below the leading edge generated the lift force of the wing. Figure 46 shows the streamlines in the fluid domain for an angle of attack of 8° with the MTE at 18°. The speed close to the walls can be seen to have been close to 0 m/s while the inlet speed was 20 m/s. The streamlines were clearly curved by the presence of the wing. The speed around the leading edge of the wing may have reached 31 m/s.

This comparison between the experimental simulated wing results shows that in order to obtain the same maximum lift coefficient, the LE surface of the simulated MLE had to be degraded compared to a perfect LE. The need for degradation of the LE of the wing model indicates that an improvement of the surface finish of the MLE could increase the impact on the stall angle by delaying it more than with the actual prototype of the MLE. Analysis of the wing using ANSYS Fluent has revealed a good correlation between the simulation and the experimental values of the lift and drag coefficients. By validating these lift and drag coefficient values for the prototype wing in the wind tunnel, our methodology for the simulation of an MLE can be applied to larger scale wings which cannot be tested in our wind tunnel, such as the wings of the UAS-S4 from Hydra Technologies.

## 7. Conclusion and Further Work

The results shown in this paper regarding analysis of MLE system behavior in a subsonic wind tunnel have shown that this system can act on the stall angle as an LE flap, and thus that an MLE is a good alternative to an LE flap. The simple mechanism of this morphing system using a servomotor makes it possible to keep constant the weight of the wing if LE flaps are already present in the wing. However, it is necessary to consider an increase in the current used by the servomotor. In order to obtain the maximum deformation of the MLE at 20 m/s, the servomotor has been found to need 8.6 W for a voltage of 7.2 V (during wind tunnel testing) with respect to 25.2 mW for a voltage of 7.2 V for a classical LE flap controlled by the same servomotor [1]. The difference in power (8.6 W versus 25.2 mW) comes from the needed morphing of the structure with the MLE while the LE flap would only need to counter the aerodynamic forces that are low for the wing. The design of the MLE presented in this paper has not been optimized yet but it is entirely functional.

The current deformation range of ±0.157 inches (±4 mm) for the MLE could be increased, and its power consumption could be reduced, through a better design. The increase of consumption due to the MLE may seem like a big disadvantage, but if the power required for a UAV is considered (~10 kW would be assumed for a wingspan of 16.4 feet (5 m)), a reduction of 1% on the needed thrust (reduction of drag) would save 100 W of power, which is much higher than the additional 8.6 W consumption by the controller.

## Figures and Tables

**Figure 1 biomimetics-04-00076-f001:**
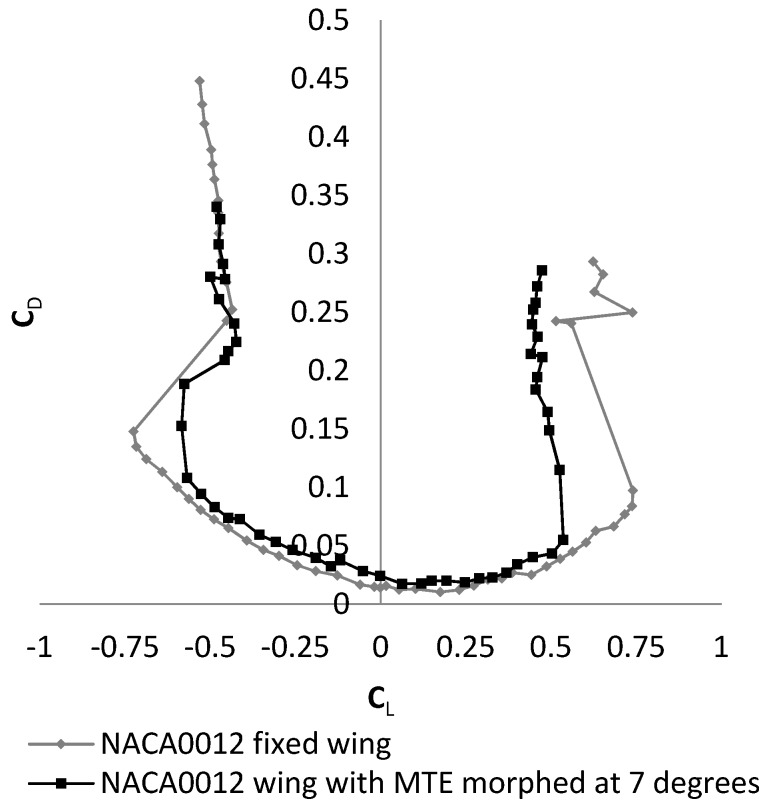
Drag coefficient variation with lift coefficient (experimental values). Legend: MTE, morphing trailing edge.

**Figure 2 biomimetics-04-00076-f002:**
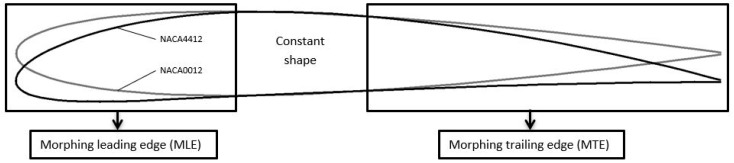
Morphing of camber: NACA0012 airfoil (grey) and NACA4412 airfoil (black).

**Figure 3 biomimetics-04-00076-f003:**
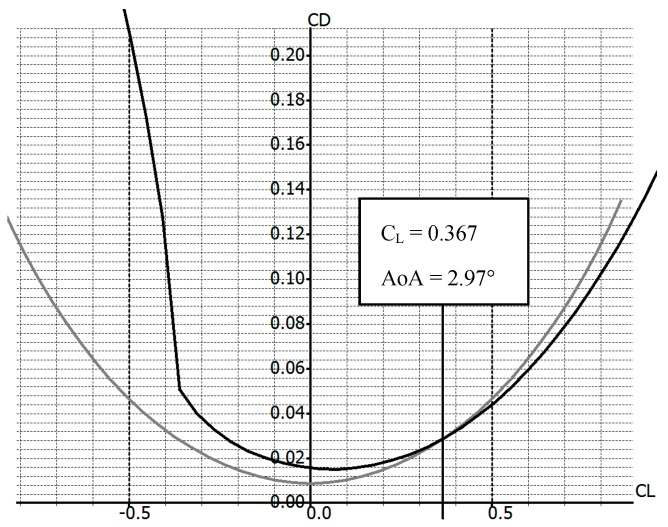
Drag coefficient variation with lift coefficient variation for NACA0012 (grey) and NACA4412 (black) (computed values). The curves intersect for a lift coefficient C_L_ = 0.367 and for an angle of attack AoA = 2.97°.

**Figure 4 biomimetics-04-00076-f004:**
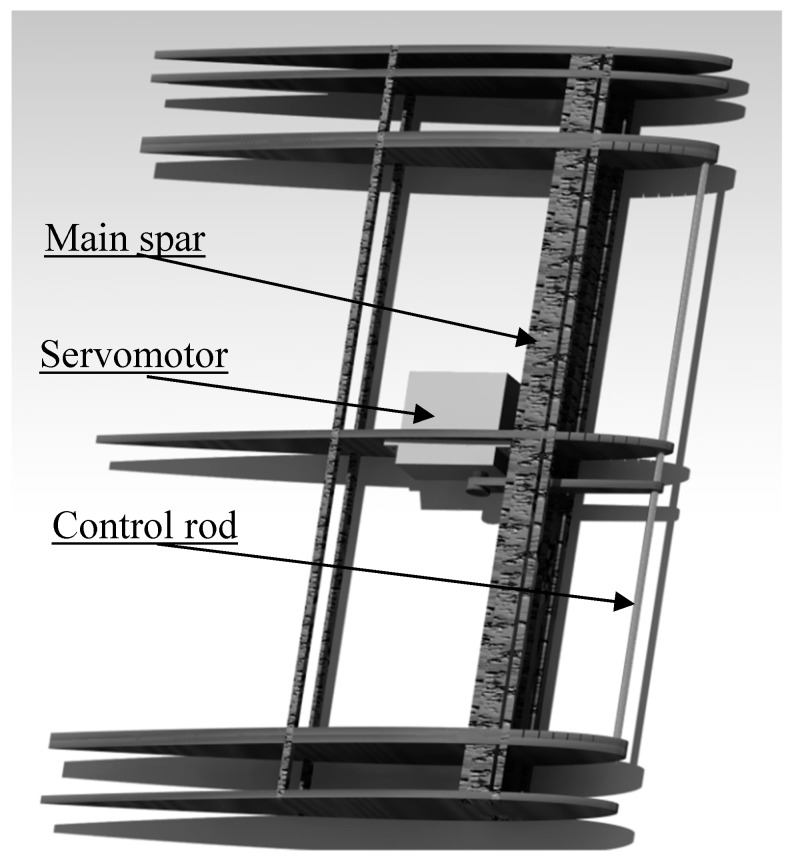
Internal structure of the wing.

**Figure 5 biomimetics-04-00076-f005:**
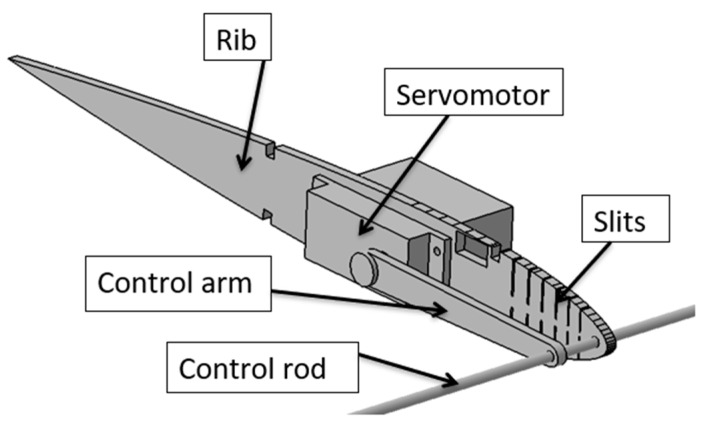
Morphing leading edge (MLE) system.

**Figure 6 biomimetics-04-00076-f006:**
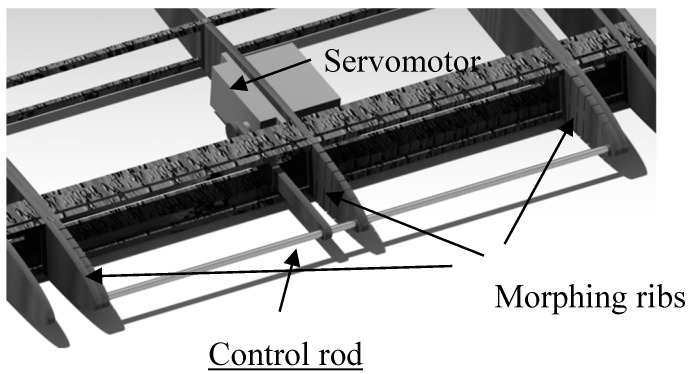
Connected morphing ribs with the control rod.

**Figure 7 biomimetics-04-00076-f007:**
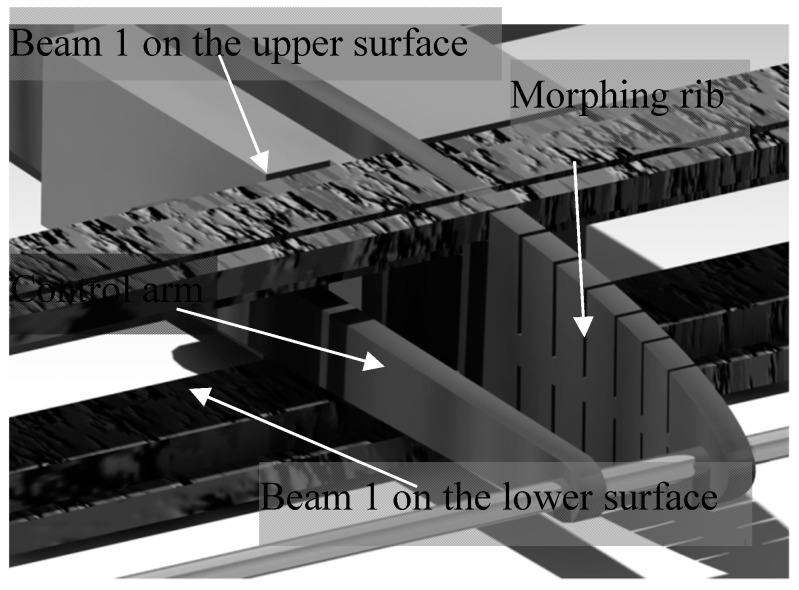
Control arm through the main spar.

**Figure 8 biomimetics-04-00076-f008:**
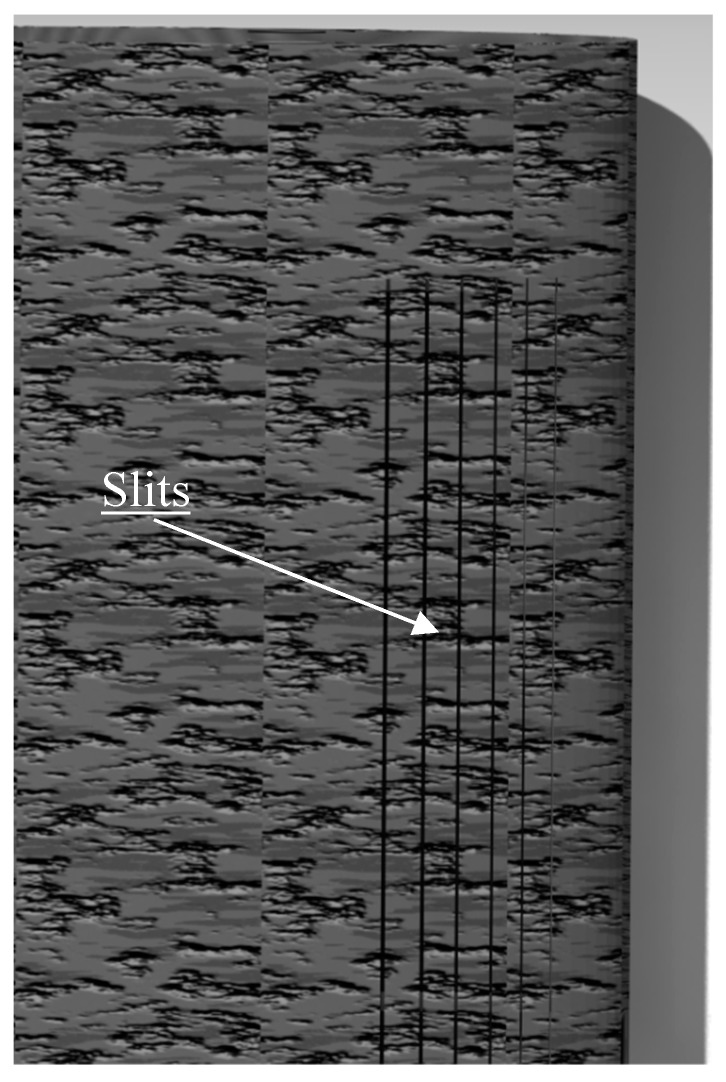
Slits on wing surface.

**Figure 9 biomimetics-04-00076-f009:**
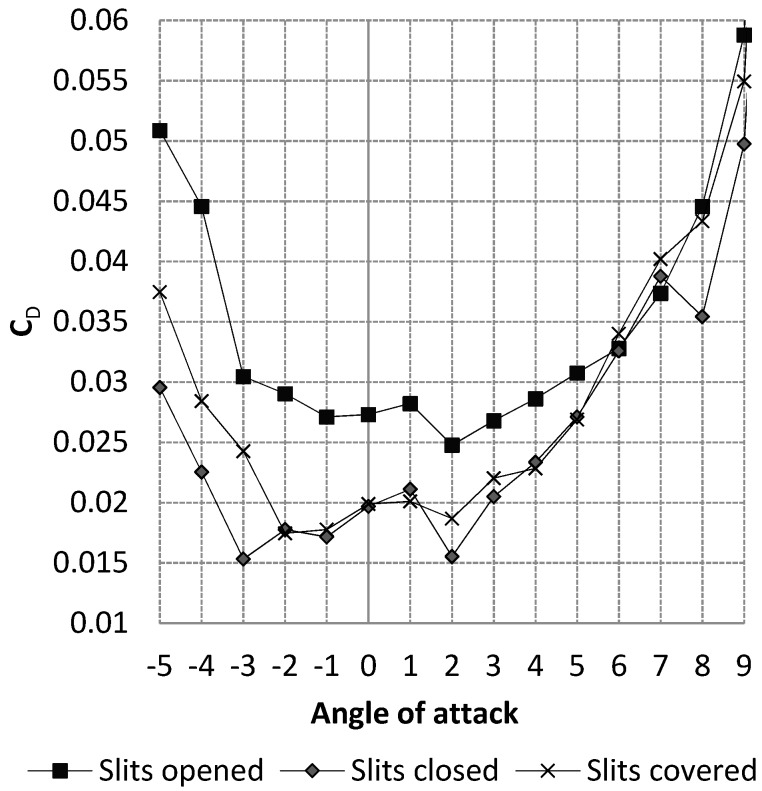
Impact of slits on wing drag.

**Figure 10 biomimetics-04-00076-f010:**
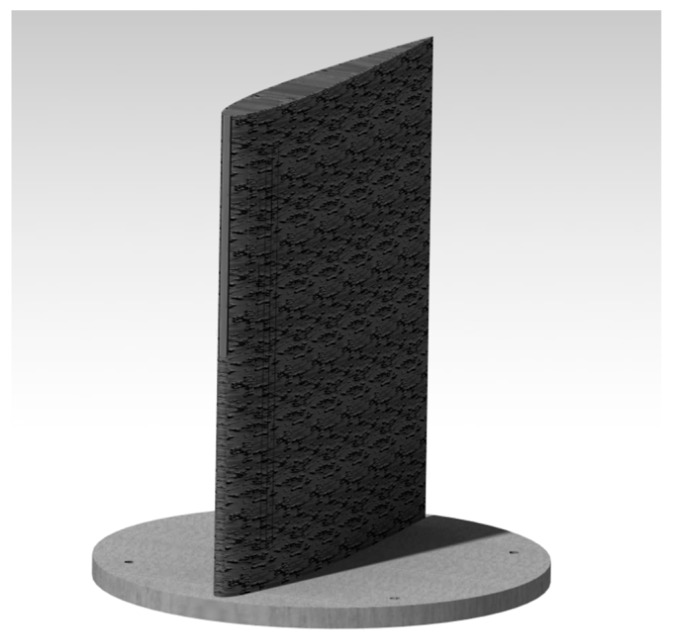
Prototype of the MLE for wind tunnel testing.

**Figure 11 biomimetics-04-00076-f011:**
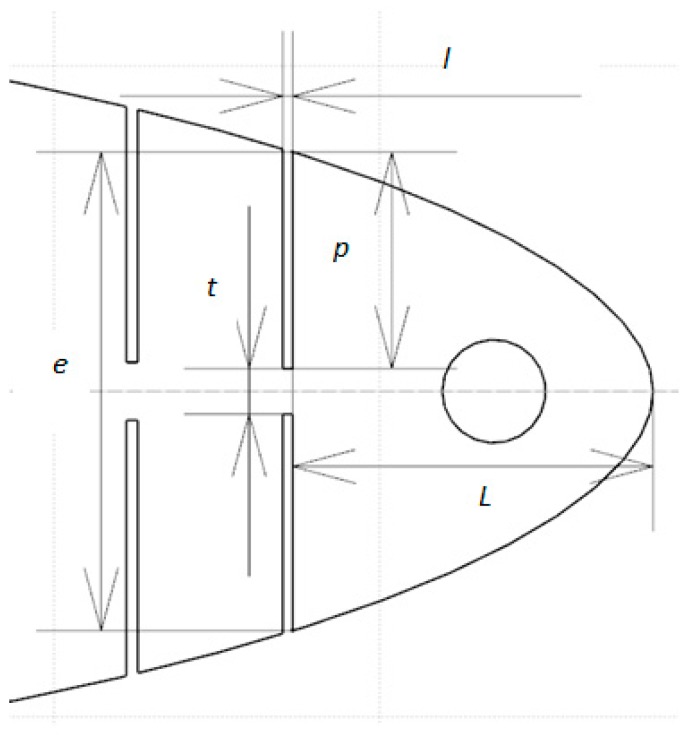
Slit parameter definitions. *l* = width of the slit, *p* = depth of the slit, *L* = distance between the slit and the LE, *e* = airfoil thickness and *t* = thickness at bottom of the slit.

**Figure 12 biomimetics-04-00076-f012:**
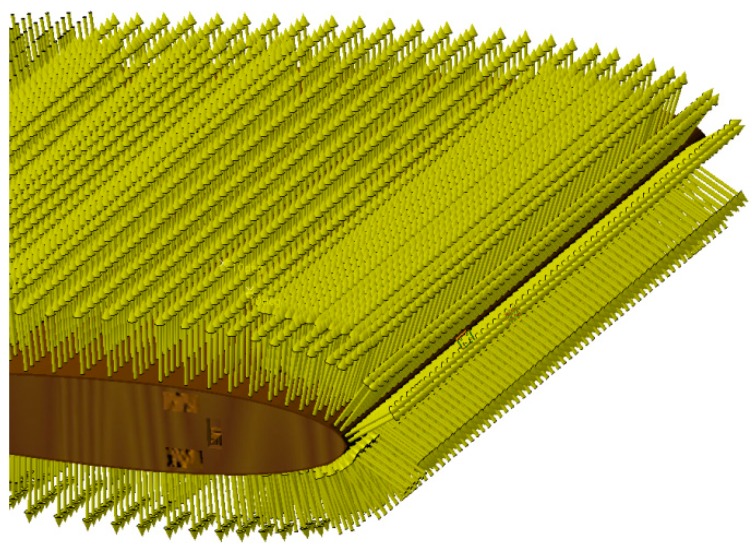
MLE with the aerodynamic loads around the wing.

**Figure 13 biomimetics-04-00076-f013:**
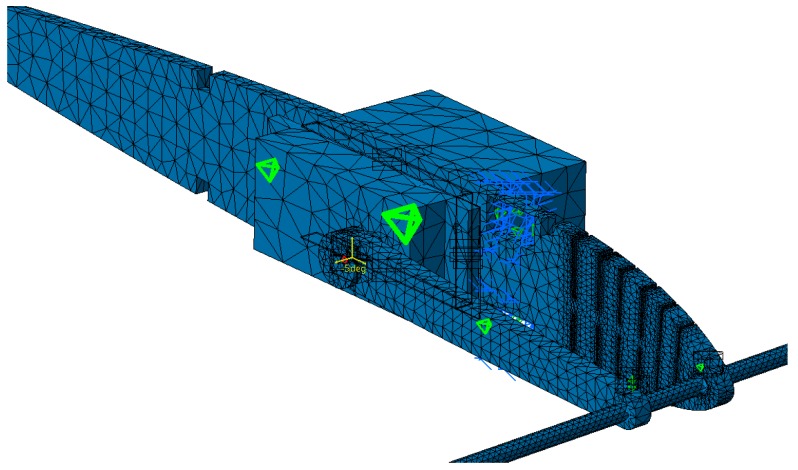
Finite element analysis (FEA) structural mesh of the MLE system.

**Figure 14 biomimetics-04-00076-f014:**
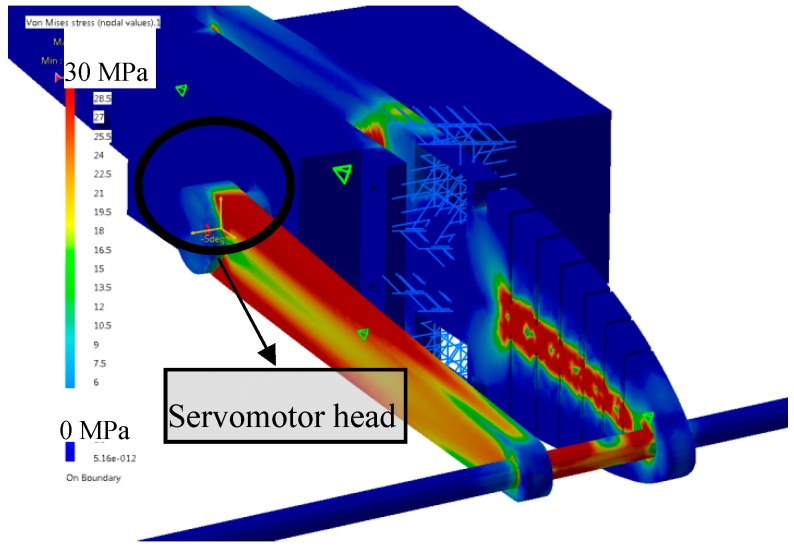
FEA of the structure of the MLE system.

**Figure 15 biomimetics-04-00076-f015:**
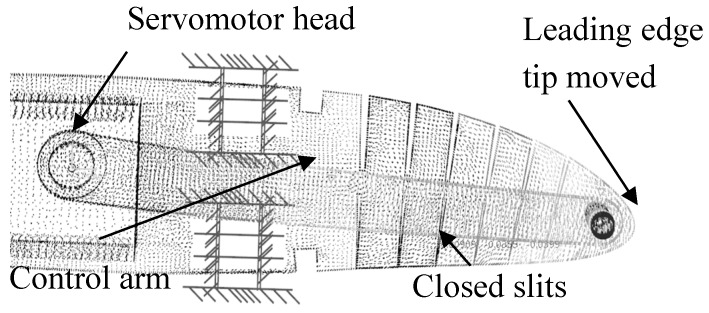
Maximum displacement of the MLE tip.

**Figure 16 biomimetics-04-00076-f016:**
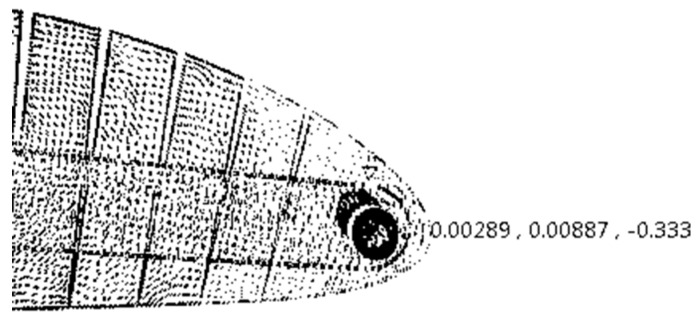
Maximum displacement values of the MLE tip.

**Figure 17 biomimetics-04-00076-f017:**
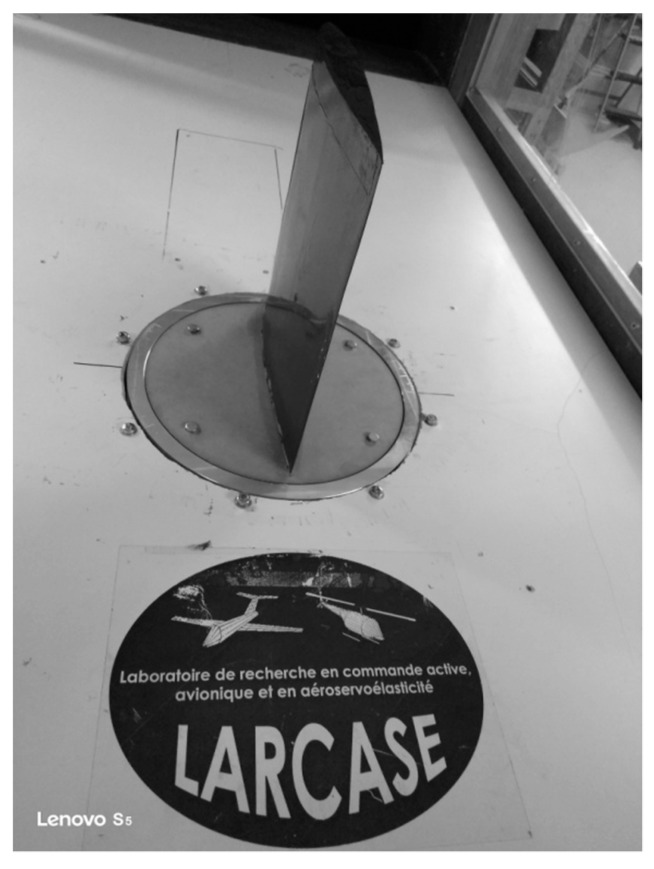
Test wing in the Price–Païdoussis subsonic wind tunnel.

**Figure 18 biomimetics-04-00076-f018:**
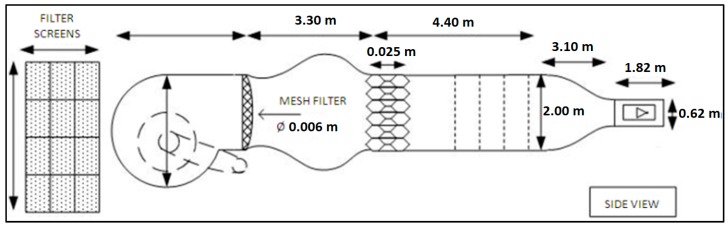
Price–Païdoussis subsonic blow down wind tunnel dimensions.

**Figure 19 biomimetics-04-00076-f019:**
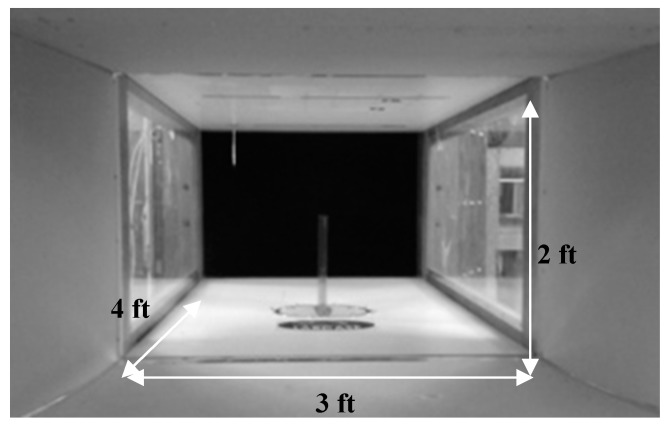
Section of the Price–Païdoussis subsonic wind tunnel of the Laboratory of Applied Research in Active Controls, Avionics, and AeroServoElasticity (LARCASE).

**Figure 20 biomimetics-04-00076-f020:**
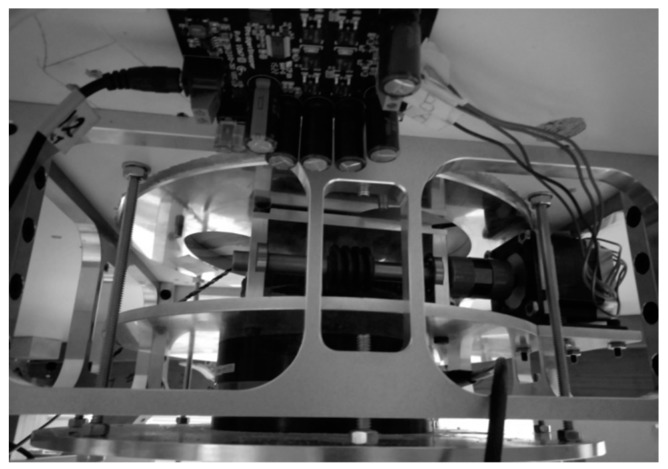
Internal mechanism of the aerodynamic loading scales.

**Figure 21 biomimetics-04-00076-f021:**
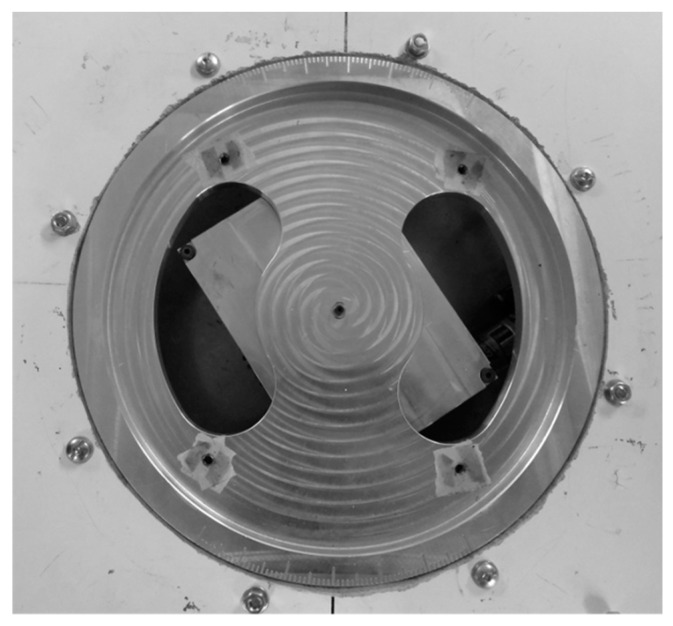
Upper disc of the aerodynamic loading scales.

**Figure 22 biomimetics-04-00076-f022:**
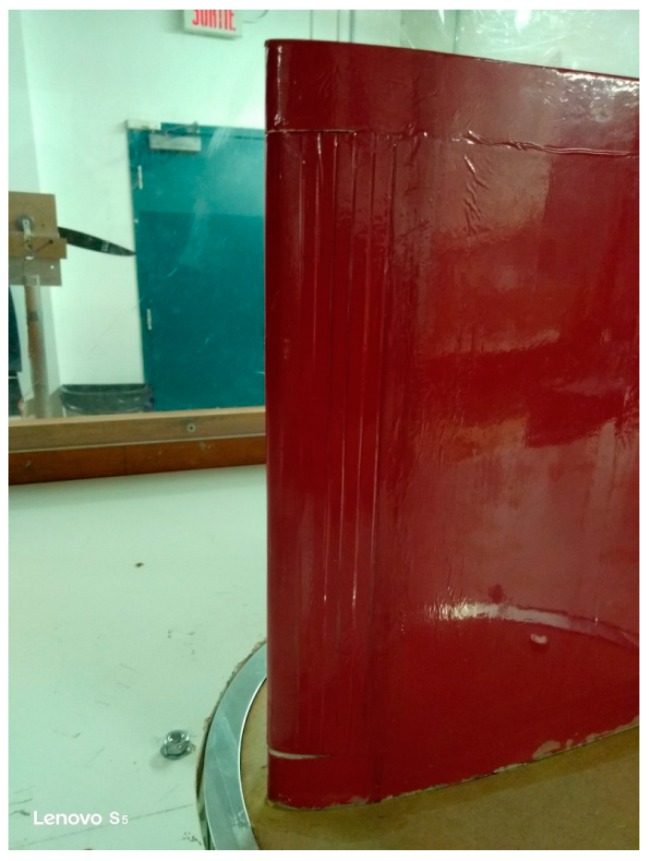
Leading edge (LE) of the prototype in its morphed configuration.

**Figure 23 biomimetics-04-00076-f023:**
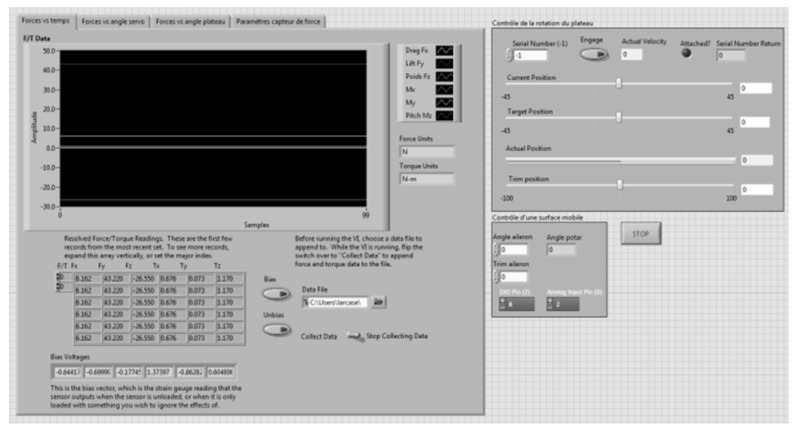
LabVIEW interface.

**Figure 24 biomimetics-04-00076-f024:**
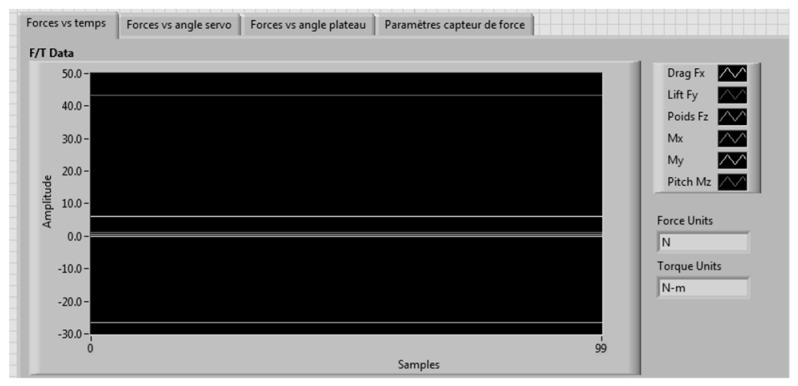
Display of forces read by F/T sensor.

**Figure 25 biomimetics-04-00076-f025:**
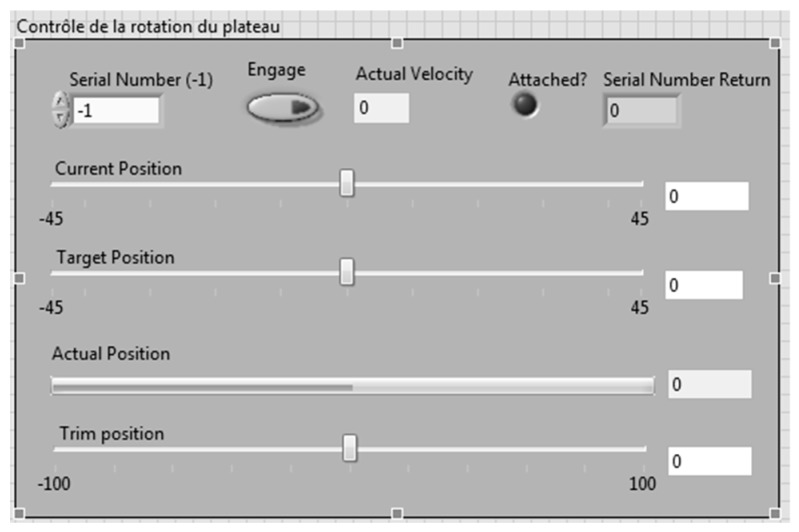
Disc angle controller.

**Figure 26 biomimetics-04-00076-f026:**
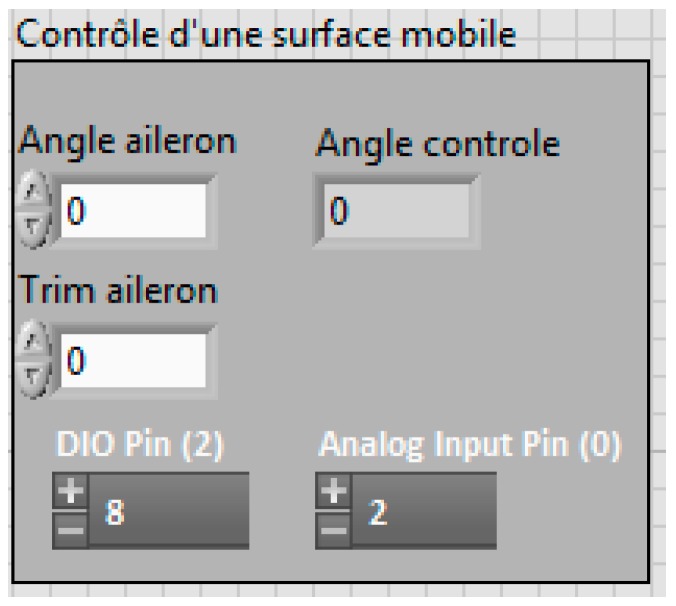
Servomotor controller.

**Figure 27 biomimetics-04-00076-f027:**
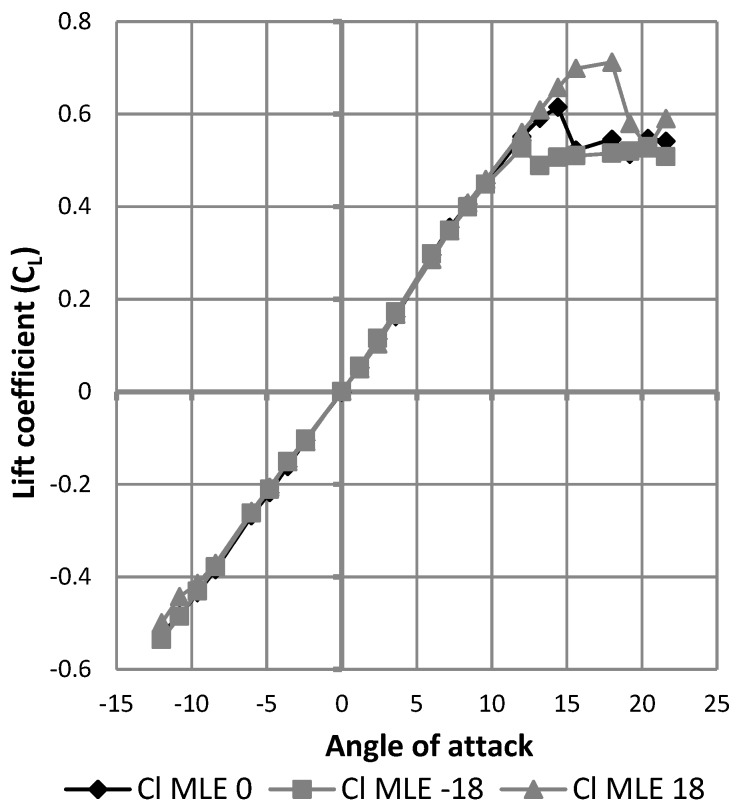
Variation of the lift coefficient with angle of attack.

**Figure 28 biomimetics-04-00076-f028:**
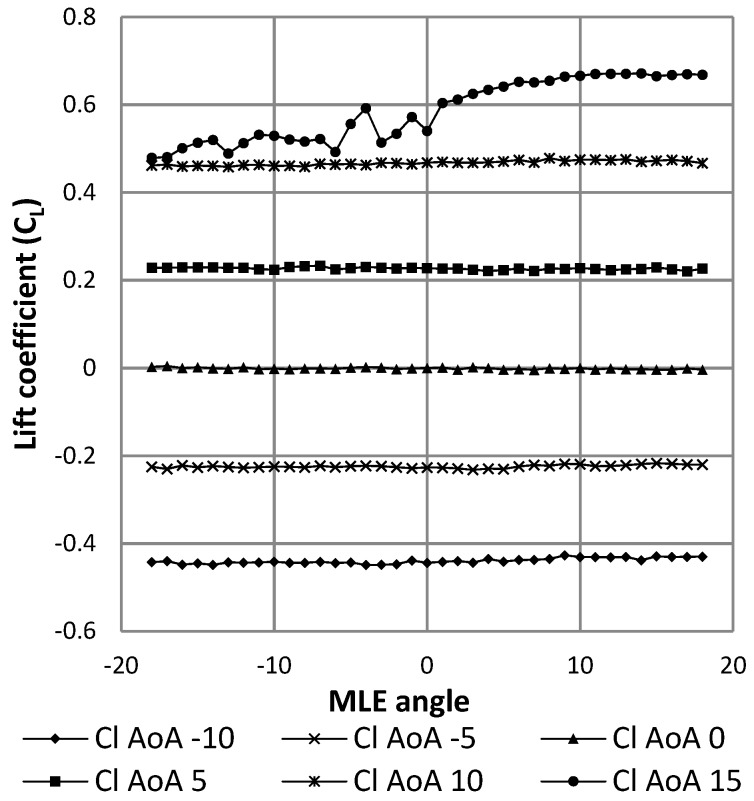
Variation of the lift coefficient with morphing angle of the MLE.

**Figure 29 biomimetics-04-00076-f029:**
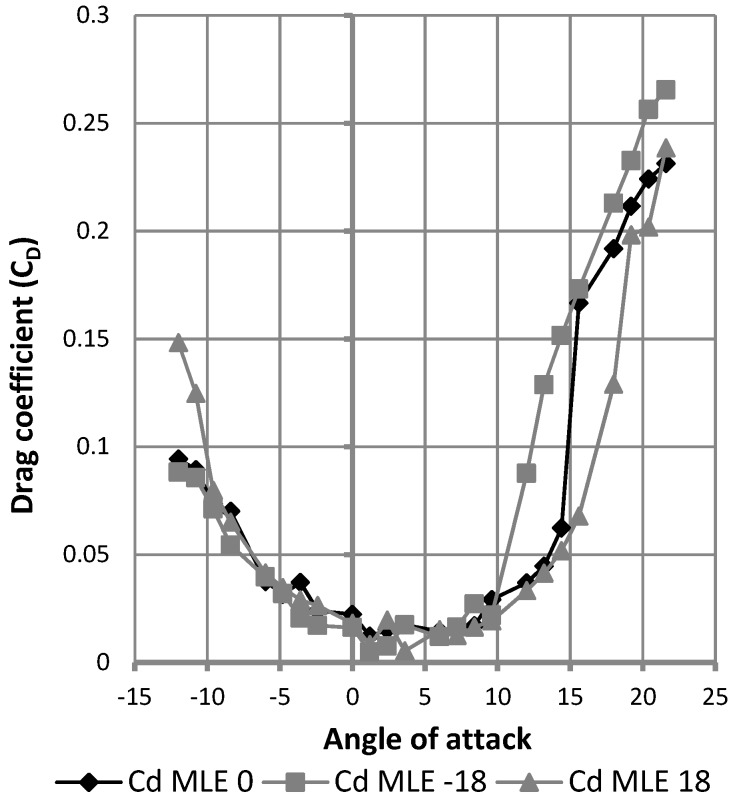
Variation of the drag coefficient with angle of attack.

**Figure 30 biomimetics-04-00076-f030:**
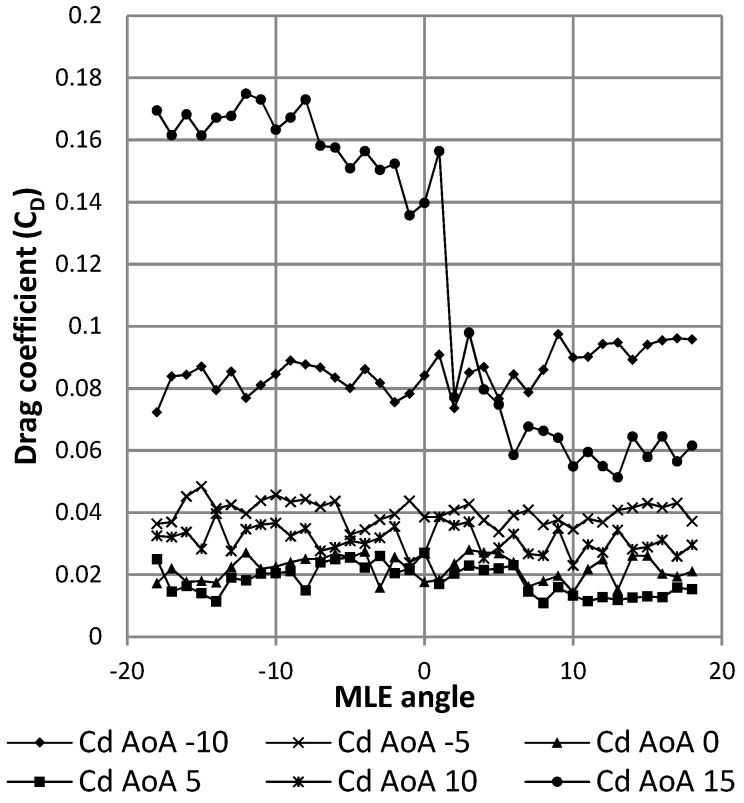
Variation of the drag coefficient with morphing angle of the MLE.

**Figure 31 biomimetics-04-00076-f031:**
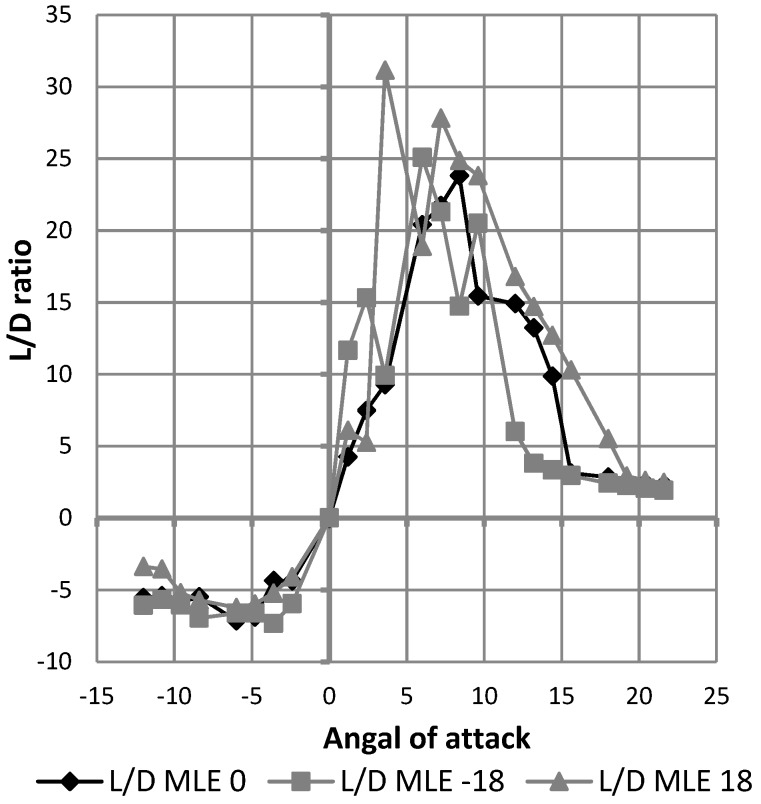
Variation of the lift on drag (L/D) ratio with angle of attack.

**Figure 32 biomimetics-04-00076-f032:**
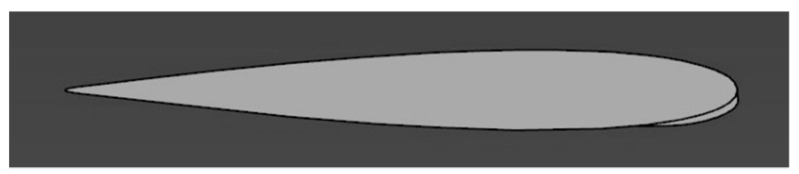
Wing with 4 mm displacement of LE tip.

**Figure 33 biomimetics-04-00076-f033:**
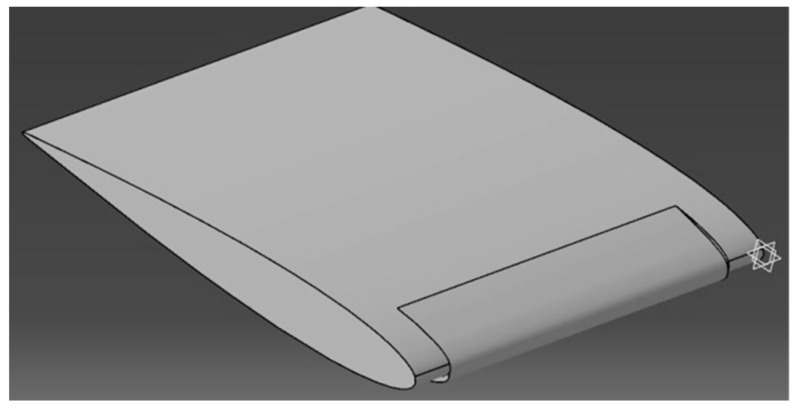
Model of wing with MLE.

**Figure 34 biomimetics-04-00076-f034:**
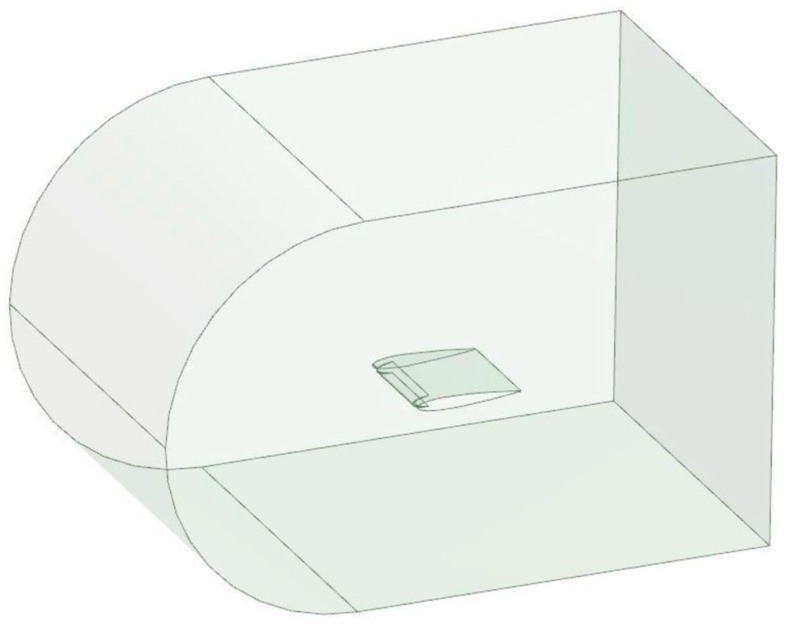
Fluid domain of the simulation.

**Figure 35 biomimetics-04-00076-f035:**
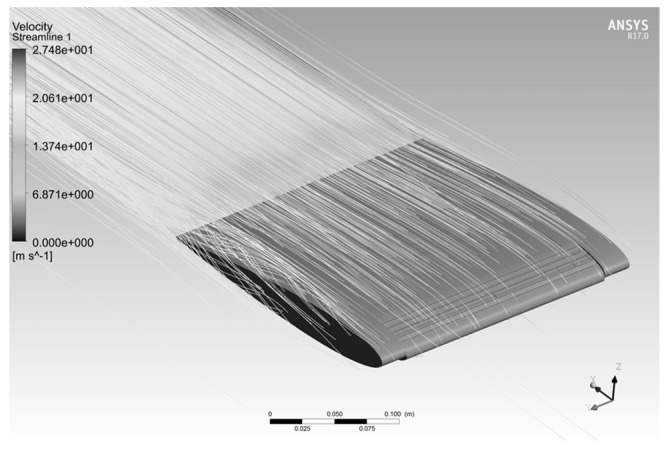
Fluent simulation.

**Figure 36 biomimetics-04-00076-f036:**
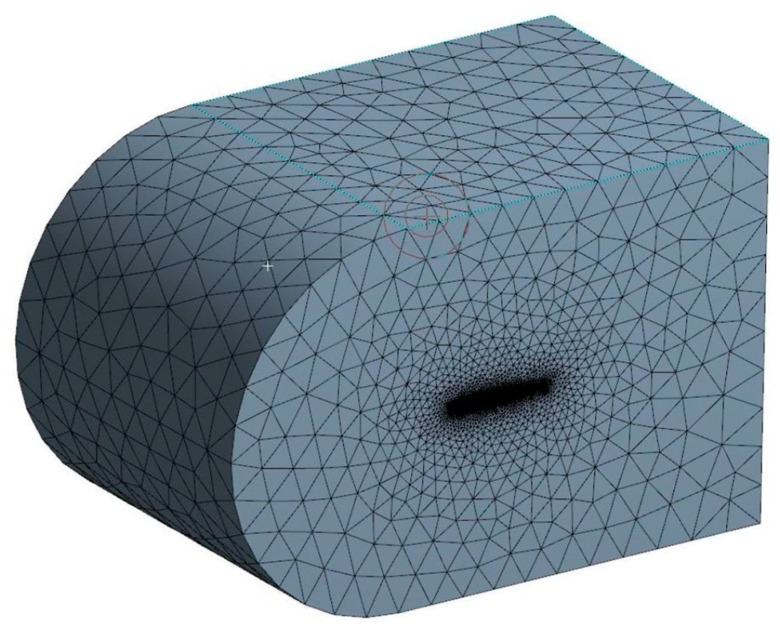
Fluid domain mesh of the simulation.

**Figure 37 biomimetics-04-00076-f037:**
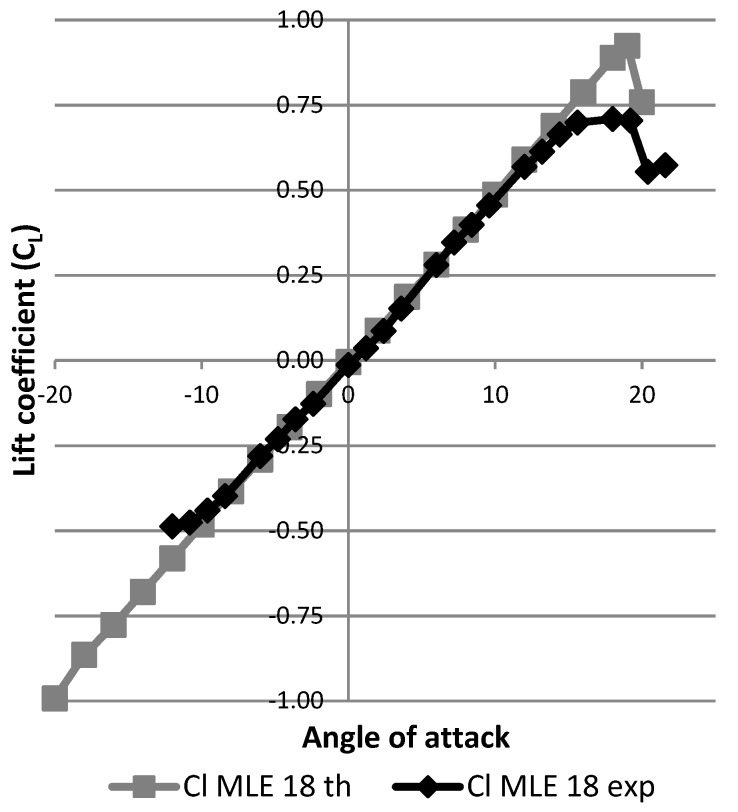
Lift coefficient variation with angle of attack.

**Figure 38 biomimetics-04-00076-f038:**
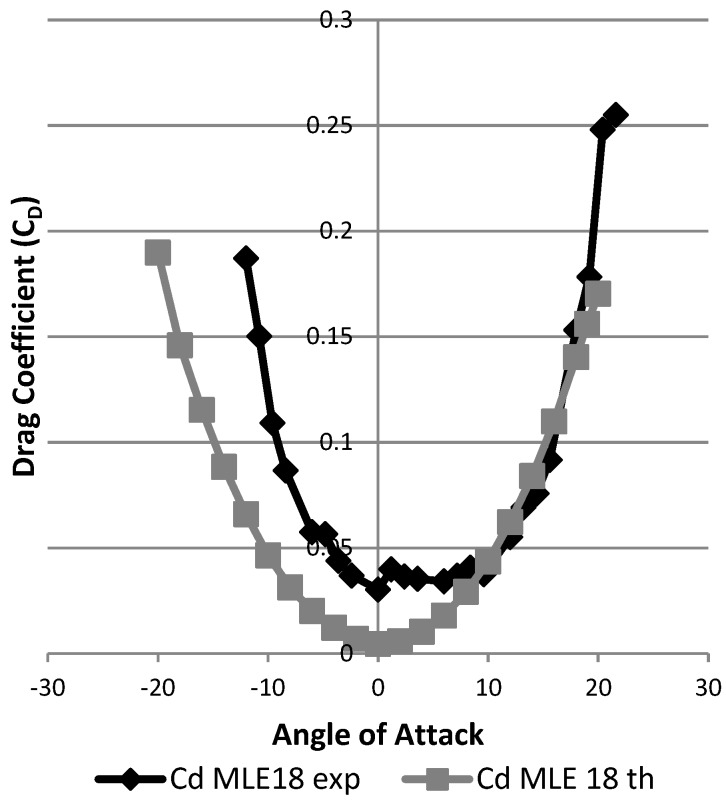
Drag coefficient variation with angle of attack.

**Figure 39 biomimetics-04-00076-f039:**
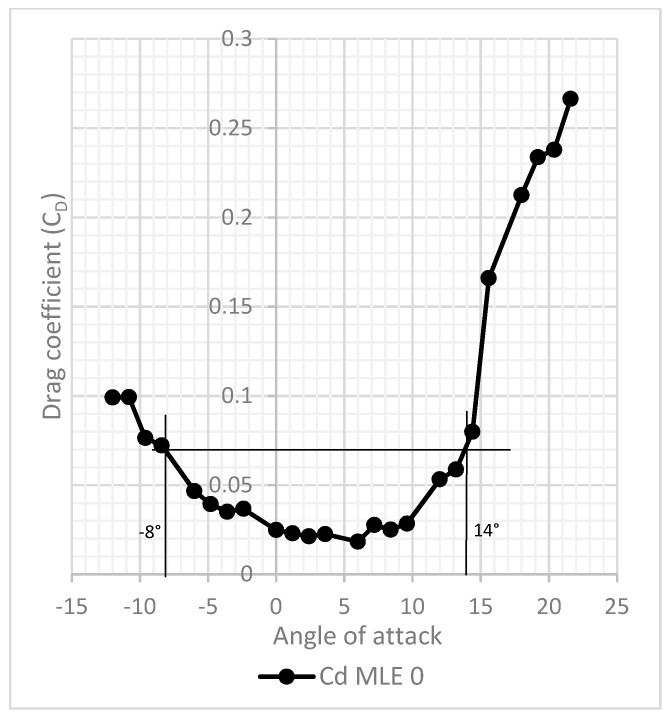
Drag variation with angle of attack for MLE at 0°.

**Figure 40 biomimetics-04-00076-f040:**
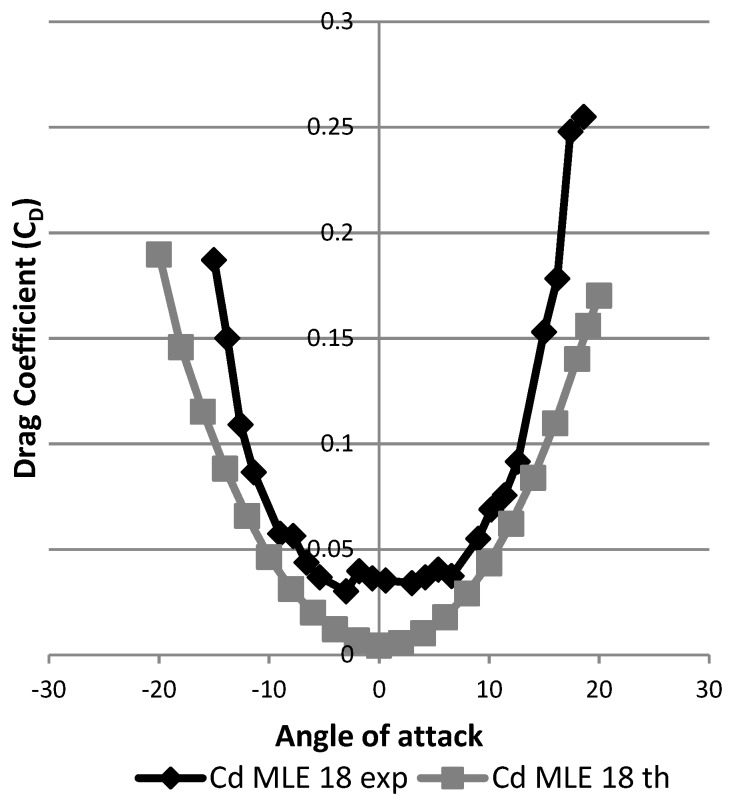
Drag coefficient variation with angle of attack with experimental values shift by 3°.

**Figure 41 biomimetics-04-00076-f041:**
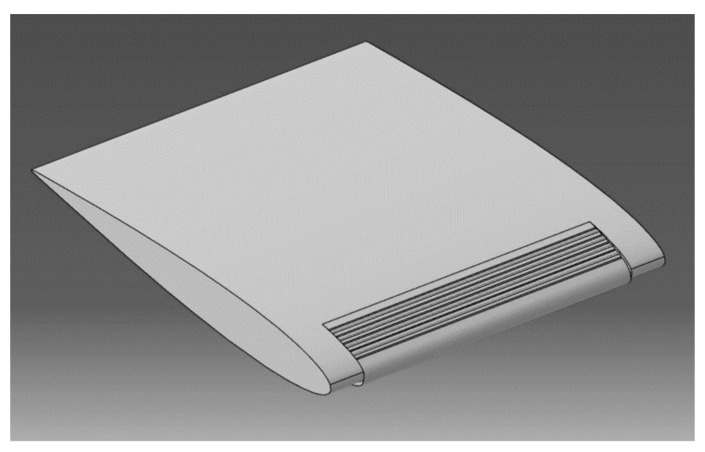
MLE with grooves and bumps.

**Figure 42 biomimetics-04-00076-f042:**
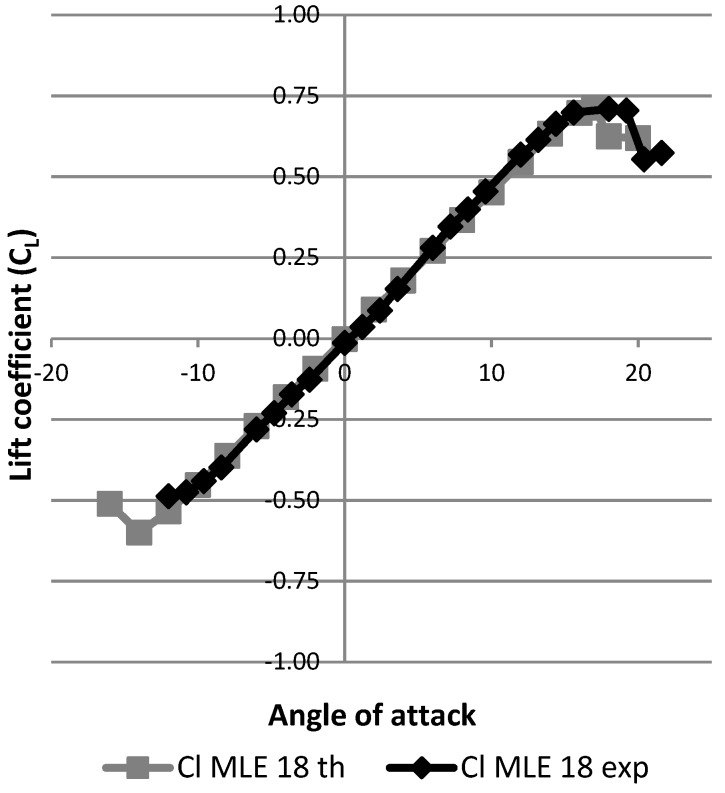
Lift coefficient variation with angle of attack for improved modeling.

**Figure 43 biomimetics-04-00076-f043:**
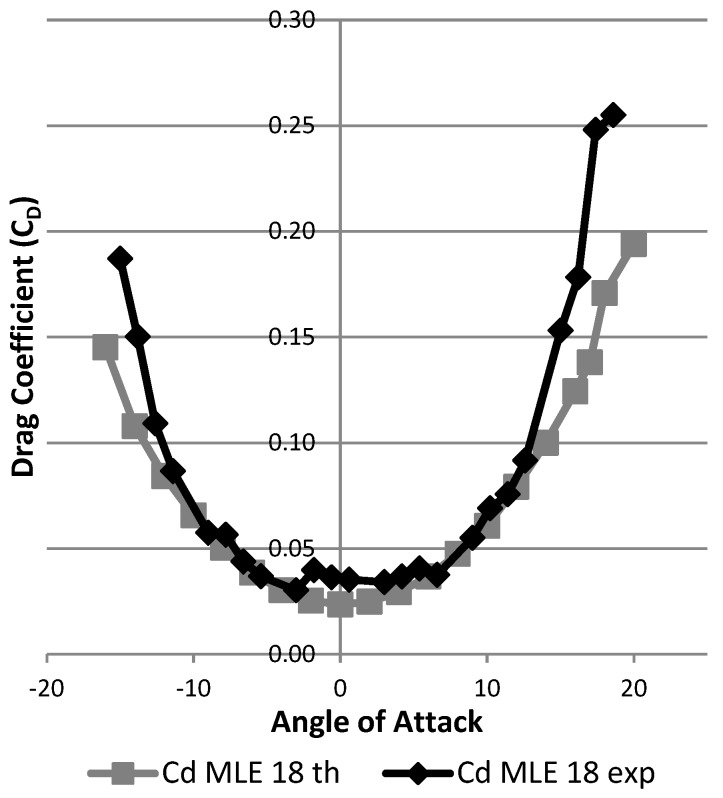
Drag coefficient variation with angle of attack for improved modeling.

**Figure 44 biomimetics-04-00076-f044:**
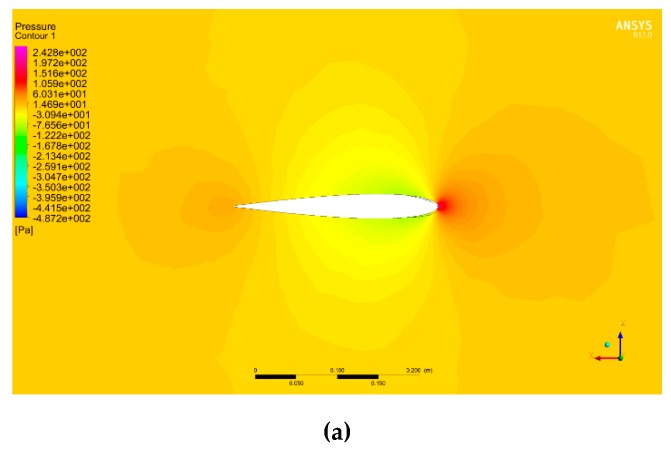
Pressure around the wing with MLE at 18° with an airspeed of 20 m/s and (**a**) angle of attack 0°, (**b**) angle of attack 10°, and (**c**) angle of attack 18°.

**Figure 45 biomimetics-04-00076-f045:**
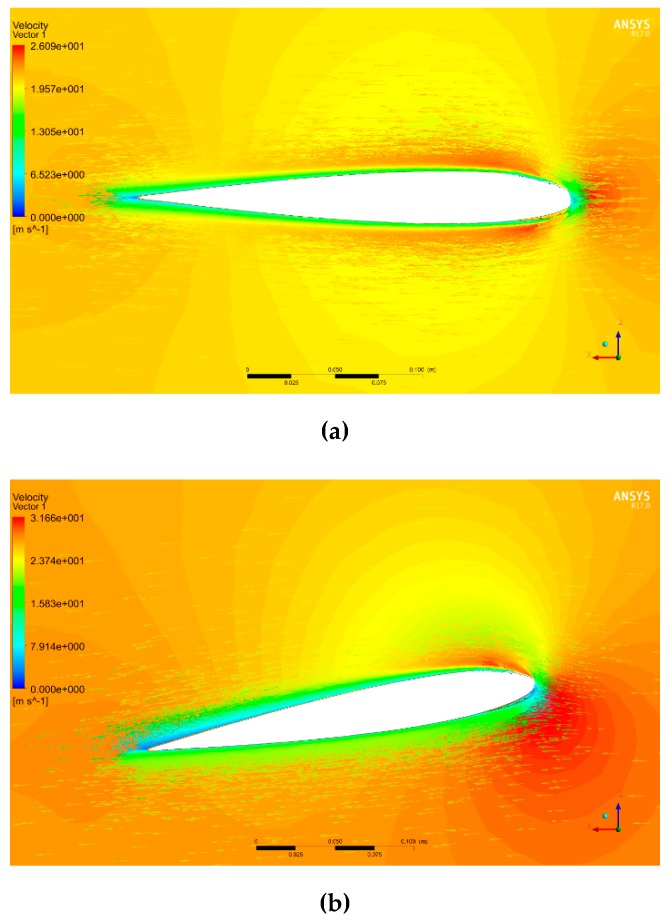
Velocity vector around the wing with MLE at 18° with an airspeed of 20 m/s and (**a**) angle of attack 0°, (**b**) angle of attack 10°, and (**c**) angle of attack 18°.

**Figure 46 biomimetics-04-00076-f046:**
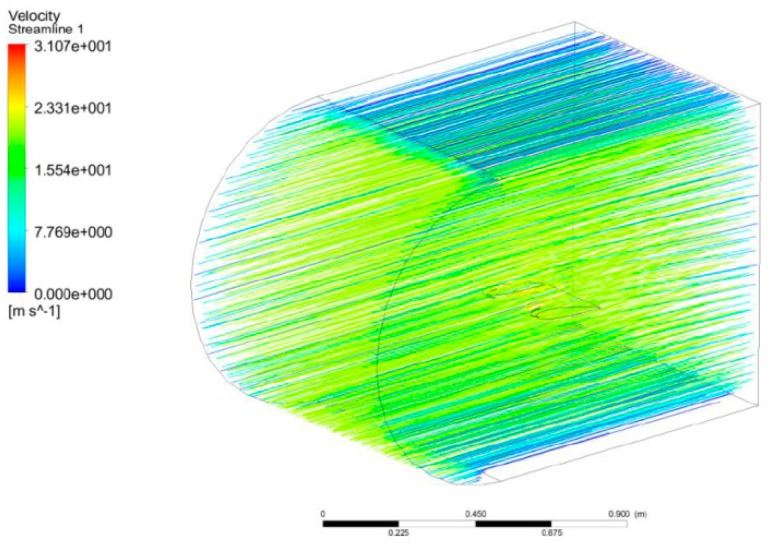
Velocity streamlines for an angle of attack of 8° and the MTE at 18°.

**Table 1 biomimetics-04-00076-t001:** Dimensions of the slits.

Slit Number	*e* (in)	*t* (in)	*l* (in)	*p* (in)	*L* (in)	*MLE* Angle (rad)	*y* (in)
1	0.579	0.055	0.012	0.262	0.436	0.046	0.020
2	0.682	0.070	0.014	0.306	0.623	0.046	0.029
3	0.765	0.085	0.018	0.340	0.817	0.053	0.043
4	0.842	0.100	0.021	0.371	1.037	0.057	0.059
5	0.902	0.115	0.024	0.394	1.258	0.061	0.077
6	0.957	0.130	0.026	0.413	1.507	0.063	0.095

**Table 2 biomimetics-04-00076-t002:** NEMA 23 bipolar stepper product specification.

Motor Type	Bipolar Stepper	Recommended Voltage	12 V DC
**Manufacturer Part Number**	57STH56-2804MB	**Rated Current**	2.8 A
**Step Angle**	0.9°	**Coil Resistance**	900 mΩ
**Step Accuracy**	±5%	**Phase Inductance**	4.5 mH
**Holding Torque**	12 kg cm	**Shaft Diameter**	1/4”
**Rated Torque**	11.2 kg cm	**Rear Shaft Diameter**	3.9 mm
**Maximum Motor Speed**	2150 RPM	**Mounting Plate Size**	NEMA23
**Acceleration at Max Speed**	80 0001/16 steps/sec²	**Weight**	695 g
**Number of Leads**	4	**Wire Length**	300 mm

**Table 3 biomimetics-04-00076-t003:** Range and resolution for ATI Omega 160 force and torque (F/T) sensor. Fx,y,z represent the forces and Tx,y,z represent the torques.

SI-1000-120 US-200-1000	Fx, Fy	Fz	Tx, Ty	Tz
**Sensing Ranges**	1000 N (200 lbf)	2500 N (500 lbf)	120 Nm 1000 lbf-in	120 Nm 1000 lbf-in
**Resolution**	1/4 N 1/32 lbf	1/4 N 1/16 lbf-in	1/40 Nm 1/8 lbf-in	1/80 Nm 1/8 lbf-in

**Table 4 biomimetics-04-00076-t004:** PhidgetStepper Bipolar HC product specification.

Motor Type	Bipolar Stepper	Available Current per Coil Max	4 A
**Number of Motor Ports**	1	**Supply Voltage Min**	10 V DC
**Motor Position Resolution**	1/16 Step (40-Bit Signed)	**Supply Voltage Max**	30 V DC
**Position Max**	±1E+15 1/16 steps	**Current Consumption Min**	25 mA
**Stepper Velocity Resolution**	1 1/16 steps/sec	**Power Jack**	5.5 × 2.1 mm Center Positive
**Stepper Velocity Max**	250,000 1/16 steps/sec	**Recommended Wire Size (Motor Terminal)**	12 to 26 AWG
**Stepper Acceleration Resolution**	1 1/16 steps/sec²	**Recommended Wire Size (Power Terminal)**	12 to 26 AWG
**Stepper Acceleration Min**	2 1/16 steps/sec²	**Operating Temperature Min**	−20 °C
**Stepper Acceleration Max**	1E+07 1/16 steps/sec²	**Operating Temperature Max**	85 °C

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
