# Peer review of "Design, Manufacturing, and Testing of a New Concept for a Morphing Leading Edge using a Subsonic Blow Down Wind Tunnel"

_biomimetics, 2019, doi:10.3390/biomimetics4040076_

Round 1

Reviewer 1 Report

This paper is about a technological system aimed at morphing the leading edge of a wing made by extruding a symmetric airfoil. The content of the article is original and of highest interest since morphing is one of the most promising way to reduce drastically gas consumption and noise. The morphing system is mounted on a wind tunnel model and tested in flows. Comparisons with numerical simulations are also carried out.

Nevertheless some improvement can be brought to the paper before its publication.

First remarks: all references to equations or figures are not available (Error Reference source not found). Furthermore the first character of numerous lines (93, 127, 163, 169, 188, 208, 217, 226, 250, 283, 302, 348,353, 361, 389, 415, 439, 446, 449) are missing.There is also a problem with the reference 27 in the bibliography part. Figure 31 is missing.

Second set of remarks:

l12: trailing or leading edge ?

Figure 20 is unclear probably due to the grey contrasts. The morphed shape can not be seen.

Figure 9: first of all, the authors presents a drag polar curve, but they should tell from where comes the results (wind tunnel tests or numerical simulations and in that case with which numerical technology and which code: inviscid-viscous flows coupling, RANS...). They also should tell for what aerodynamic conditions they got their results, probably for incompressible flows. At last an analysis of the nature of the flow could be valuable. Indeed, if the flow is laminar on a large part of the chord, the drag will be strongly influenced by the transition, especially if the slits or humps trigger the transition. At very last, be careful with the drag scale which is very large. One interval matches 20 drag count. So a quick analysis is required here.

Table 1: it would be interesting to have the chord in order to have a reference value, since the parameters are given in English units.

I am not sure that the long part dedicated to the LabVIEW interface is worth and brings valuable elements to what the authors want to demonstrate (from line 249 to 273). This part could at least be greatly shortened.

Line 291: the authors make a comparison with a "classical slat", but they never show results obtained with a conventional slat. Such comparisons would bring a great plus to the demonstration of the interest of a MLE.

Line 328, 329 and 331: there is an inconsistency with the figures in inches and mm.

Section 6: Some questions arise about the mesh. First, what about the boundary conditions: far field or is the wind tunnel test section modeled? The description of the mesh characteristics are unclear for readers that do not use Fluent (from line 359 to 372). "proximity and curvature size function"?, "relevance of -10" ?, "transition ratio" ?, "face sizing" ?. Nevertheless, a boundary layer modeled with only 5 layers is obviously much too coarse to get accurate results in particular for high incidences (close to stall). A growth rate of 1.2 seems also too large. Care must be taken with wall functions: the user has to check that y+ is actually about 100, and they are not well suited for separated flows occurring for high incidences especially close to stall. Such aerodynamic conditions remain today very challenging for CFD and RANS simulations. The flow separations are very difficult to predict using classical 2 equations turbulence models.

l383: a deeper analysis of the experimental results could help to explain why there is a shift of 3 deg for the drag curve to recover symmetry (due to the experimental setup?). Furthermore the LE deflection should break this symmetry and should thus have an impact on the drag curve. That is why the superpositions of both numerical and experimental curves for LE deflection of both 0deg and 18deg would be interesting.

The term "theoretical" would not be the most suited to refer to the results from the numerical simulations. "numerical" would be better.

The numerical modeling of the non smooth surface is very interesting. The authors may add the characteristics of the grooves and bumps taken into account by the mesh. It would be especially interesting to know if this bumps have been built from measurement of the experimental model or if they are generic ones.

Figure 39: negative pressure is always surprising. so the reference p0 should be given. Same scales would help the reader for the figures analysis.

Figure 40 same remark for the scales. The authors should also add what velocity component is drawn, unless it is the velocity norm. Figure c is surprising since he separations are not easy to see. But I have some doubts about the validity of RANS-CFD for post stall incidences. Drawing stream lines could help.

Conclusions:

The comparisons between the energy consumption of the MLE and of a conventional LE flap is a valuable added value of the paper. But The authors should add precisions about how they have computed those energy consumptions, and most of all about how they have converted them for the same aircraft in order to compare them.

I recommend the publication of this article but only after minor modifications that take into account the first set of remarks. I invite the authors to take the opportunity of this minor revision to take into account the second set of remarks.

Author Response

We would like to thank very much to reviewer 1 for its comments. Please find below the comments of reviewer, and for each comment, our answers are given below.

This paper is about a technological system aimed at morphing the leading edge of a wing made by extruding a symmetric airfoil. The content of the article is original and of highest interest since morphing is one of the most promising way to reduce drastically gas consumption and noise. The morphing system is mounted on a wind tunnel model and tested in flows. Comparisons with numerical simulations are also carried out.

Nevertheless some improvement can be brought to the paper before its publication.

Comment 1.1:

First remarks: all references to equations or figures are not available (Error Reference source not found). Furthermore, the first character of numerous lines (93, 127, 163, 169, 188, 208, 217, 226, 250, 283, 302, 348,353, 361, 389, 415, 439, 446, 449) are missing. There is also a problem with the reference 27 in the bibliography part. Figure 31 is missing.

Answer 1.1:

These errors were not in the original manuscript. We believe that they were introduced accidentally during the submission process. We have done all corrections, including ref. 27 and the renumbering of figures. Please accept our apologies

Second set of remarks:

Comment 1.2:

l12: trailing or leading edge?

Answer 1.2:

L13: This sentence was removed following a comment by the reviewer 2 as it referred to a previous article.

Comment 1.3:

Figure 20 is unclear probably due to the grey contrasts. The morphed shape can not be seen.

Answer 1.3:

Morphed shape is hard to see on Figure 20 (now Figure 22) as the displacement is very small. This figure has been now represented in colors to improve its understanding. The morphed shape is not seen, but the difference of shapes between the fixed and the morphing parts of the leading edge are shown.

L278: We also added a presentation of the figure: “Figure 22 show the leading edge of the MLE where the difference of shapes between the fixed and the morphing part of the leading edge can be seen at the root of the wing.”

Comment 1.4:

Figure 9: first of all, the authors presents a drag polar curve, but they should tell from where comes the results (wind tunnel tests or numerical simulations and in that case with which numerical technology and which code: inviscid-viscous flows coupling, RANS...).
They also should tell for what aerodynamic conditions they got their results, probably for incompressible flows. At last an analysis of the nature of the flow could be valuable.
Indeed, if the flow is laminar on a large part of the chord, the drag will be strongly influenced by the transition, especially if the slits or humps trigger the transition. Finally, we need to be careful with the drag scale which is very large. One interval matches 20 drag count. Therefore, a quick analysis is required here.

Answer 1.4:

The drag curve was obtained experimentally using wind tunnel tests where flow was considered incompressible. The following sentence was added: L129: “The values presented in Figure 9 were obtained during wind tunnel tests on the MTE at a speed of 15 m/s [1].”

The drag scale in Figure 9 has been changed to an interval of “5 drag count” instead of “20 drag count”.

Comment 1.5:

Table 1: it would be interesting to have the chord in order to have a reference value, since the parameters are given in English units.

Answer 1.5:

The value of the chord is added on line 203:

“These dimensions correspond to a “reference” wing chord of 10 in (25.4 cm).”

Comment 1.6:

I am not sure that the long part dedicated to the LabVIEW interface is worth and brings valuable elements to what the authors want to demonstrate (from line 249 to 273). This part could at least be greatly shortened.

Answer 1.6:

The LabVIEW interface is explained in the experimental setup section because of the fact that it was developed for the wind tunnel tests presented in this article.

Comment 1.7:

Line 291: the authors make a comparison with a "classical slat", but they never show results obtained with a conventional slat. Such comparisons would bring a great plus to the demonstration of the interest of a MLE.

Answer 1.7:

Unfortunately, we do not have a readily practical demonstrator model with a “classical slat”. This model will be done in a next research step in order quantify the difference between result obtained with a MLE and a classical slat. The primary objective was to validate that deformation was possible, and that this deformation had an impact on the stall angle of the wing. For the behavior of a classical slat, the results of the paper “Effect of Flap and Slat Riggings on 2-D High-Lift Aerodynamics” was considered.

Comment 1.8:

Line 328, 329 and 331: there is an inconsistency with the figures in inches and mm.

Answer 1.8:

L376: It is true that there is an inconsistency in the dimensions, therefore the value of 0.157 in. was corrected to 0.313 in. for the maximum displacement of the MLE (ribs alone).
L379: When converting 0.322 in from Imperial to SI units, the precision was changed from 8 mm to 8.18 mm.

Comment 1.9:

Section 6: Some questions arise about the mesh. First, what about the boundary conditions: far field or is the wind tunnel test section modeled?

The description of the mesh characteristics are unclear for readers that do not use Fluent (from line 359 to 372). "proximity and curvature size function"?, "relevance of -10" ?, "transition ratio" ?, "face sizing" ?. Nevertheless, a boundary layer modeled with only 5 layers is obviously much too coarse to get accurate results in particular for high incidences (close to stall). A growth rate of 1.2 seems also too large.

Care must be taken with wall functions: the user has to check that y+ is actually about 100, and they are not well suited for separated flows occurring for high incidences especially close to stall. Such aerodynamic conditions remain today very challenging for CFD and RANS simulations.

The flow separations are very difficult to predict using classical 2 equations turbulence models.

Answer 1.9:

Following comment 1.9 of the reviewer, next changes were done in the paper:

L404: “The fluid domain used for the simulation corresponds to the fluid in the test section of the wind tunnel. It was generated with the DesignModeler of Ansys Fluent. The inlet has been rounded in order to reduce the volume of the fluid domain, and to reduce the computation time.”

Most of the parameters in the analysis are set as “default” values. The mesh has been determined to obtain the fastest possible computation while ensuring the results convergence. We changed the description of the mesh, and we added Figure 36 representing the mesh of the fluid domain.
L418: “The mesh in Figure 36 has been designed using the Meshing tool of Ansys Fluent. The size function proximity and curvature was used in order to obtain a very good resolution around the wing (relevance of -10). And an inflation around the wing was used in order to get the finest mesh at the surface of the wing (transition ratio of 0.272, 5 layers and growth rate of 1.2). Finally, a face sizing around the sharp corners (trailing edge, slits and bumps) was used, in order to avoid convergence problems using an element size of 1 mm For this first simulation, the mesh had around 3 652 638 elements with a good quality (average orthogonality of 0.85 and average skewness of 0.23). Using this configuration, the mesh size was calculated in 10 to 15 minutes depending on the computer execution speed.”

Indeed, it was difficult to obtain the Cl results for the stall angle measured during wind tunnel tests. This correspondence works between theorical and experimental for the displacement of the leading edge of 4 mm. But it should be tested and analyzed for other stall angles cases of deformation.

The two equations model was chosen because we were limited in our computing capabilities. We added a justification on the selection of a 2 equations model.
L442: “The 2 equations model was chosen due the computing capabilities of our computers.”

Comment 1.10:

l383: a deeper analysis of the experimental results could help to explain why there is a shift of 3 deg for the drag curve to recover symmetry (due to the experimental setup?). Furthermore the LE deflection should break this symmetry and should thus have an impact on the drag curve. That is why the superpositions of both numerical and experimental curves for LE deflection of both 0deg and 18deg would be interesting.

Answer 1.10:

The shift of 3° for the drag curve is probably due to the last structural change of the aerodynamic scales because it was not there before. The shifting is present for the other models that we tested in the wind tunnel lab following the last structural of the aerodynamic scales.

The next sentence is added:
L453: “The shifting of the drag curve is probably due to the experimental setup, but the exact cause needs to be still determined.”

The superposition of experimental curves for LE deflection of 0° and 18° is shown in Figure 29, but we can’t do the same superposition for the numerical curves because the deflection of 0° has not been computed. In Figure 29, we can see that the deflection had a very little impact on the symmetry of the curve compare with the shift of 3°.

Comment 1.11:

The term "theoretical" would not be the most suited to refer to the results from the numerical simulations. "numerical" would be better.

Answer 1.11:

L450, 457, 483 and L484: “Theoretical” was changed to “numerical” everywhere in the paper.

Comment 1.12:

The numerical modeling of the non smooth surface is very interesting. The authors may add the characteristics of the grooves and bumps taken into account by the mesh. It would be especially interesting to know if this bumps have been built from measurement of the experimental model or if they are generic ones.

Figure 39: negative pressure is always surprising. so, the reference p0 should be given. Same scales would help the reader for the figures analysis.

Figure 40 same remark for the scales. The authors should also add what velocity component is drawn, unless it is the velocity norm.

Figure c is surprising since the separations are not easy to see. But I have some doubts about the validity of RANS-CFD for post stall incidences.

Drawing stream lines could help.

Answer 1.12:

Details were added on the shapes of the grooves and bumps, their quantities and the meshes around them.
L471: “The grooves and bumps were defined using a half-circle profile with a radius of 0.5 mm. There were 5 of each on the inner surface and 5 of each on the upper surface and gave a total of 10 grooves and 10 bumps. The sizes of the mesh elements around the grooves was fixed at 0.25 mm. The sizes of the mesh elements around the rest of the wing was 1 mm.”

We add the value of the reference p0 in the description of the Figure 44 (old Figure 40).
L511: “Figure 44 shows the pressure around the wing in Pascal units for angles of attack of 0°, 10° and 18°, with the MLE at 18°, where the airspeed is 20 m/s and the reference pressure p0 is of 101 325 Pa.”

The arrow display represents the norm of the velocity vectors. A precision was added in the paper.
L518: “Figure 45 shows the norm of the velocity vector around the wing in m/s for angles of attack of 0°, 10° and 18°, with the MLE at 18°, where the airspeed is 20 m/s.”

At 18° the wing in the wind tunnel has not stalled. Therefore, Figure c does not correspond to a post stall incidence but it the angle of attack just before the stall incidence.

We add a figure with the streamline in the fluid domain.
L523: “Figure 46 shows the streamlines in the fluid domain for an angle of attack of 8° with the MTE at 18°. The speed close to the walls is close to 0 m/s while the inlet speed is 20 m/s. The streamlines are clearly curved by the presence of the wing. The speed around the leading edge of the wing may reach 31 m/s.”

Conclusions:

Comment 1.13:

The comparisons between the energy consumption of the MLE and of a conventional LE flap is a valuable added value of the paper. But The authors should add precisions about how they have computed those energy consumptions, and most of all about how they have converted them for the same aircraft in order to compare them.

Answer 1.13:

The values of energy consumption correspond to those that were measured during the wind tunnel tests. For their use by the UAV, we considered the consumption of a UAV of 5-meter wingspan.

L547: “the servomotor needed 8.6 W for a voltage of 7.2 V (during wind tunnel test)”

L554: “The increase of consumption due to the MLE may seem like a big disadvantage but if you consider the power required for a UAV (~10 kW would be considered for a wingspan of 16.4 ft (5 m))”

I recommend the publication of this article but only after minor modifications that take into account the first set of remarks. I invite the authors to take the opportunity of this minor revision to take into account the second set of remarks.

Many thanks once again to reviewer 1 for his comments.

Reviewer 2 Report

The presented morphing mechanism is a variant of different shape adaptable leading edge concepts already proposed in literature. The most critical issue of this kind of mechanisms is the deformable skin. The authors didn't address this aspect neither they proposed a valid solution. Moreover the manuscript contains some interesting results, but together with many other things that are presented in a confusing and incosistent way. For all these reasons a deep review of the whole manuscript is needed to publish it in a scientific journal.
The deep review should include the following issues providing a meticulous clarification.

- First of all, using the name of the Price-Paidoussis Wind Tunnel in the title of a scientific is not appropriate. Moreover, the first piece of the introduction seems to be dedicated to sponsor Price-Paidoussis Wind Tunnel, while it should be used to present a method or a system, for example, in general.

- In the first sentence of the abstract authors wrote about 'this' morphing leading edge system before they introduce it. I suggest to replace 'this' whit 'a' generic system that is still to be presented. Moreover the following sentence should be removed 'the first part of this research concerns the design of a morphing trailing edge', as well as no results should be presented in the abstract.

- The paper is full of messages like 'Error! Reference source not found' that make manuscript reading very difficult.

- The first general aspect that must be clarified throughout the manuscript is that the proposed morphing concept can be only used as a wind tunnel model. It cannot be proposed as a general system suitable to be applied to the aircraft design. This because it is not able to bear the internal forces typical of an actual wing, such as the torsion (there is not a structural skin), and the external surface has many defects required by the system to work, but that would not be acceptable from the aerodynamic point of view on an actual aicraft.
Despite this, it seems that the authors propose this morphing mechanism as a general concept for all the aeronautical community. They wrote about the use of materials and actuators common in aircraft design and, again, about the prototype that was designed to validate the functionality of the deformation method, so the concept is general?
In my opinion the presented mechanism can be valid only to manufacture a wind tunnel model and to perform some aerodynamic test.

- Another general comment that must be clarified is related to the structural properties of the MLE. The authors wrote that to facilitate the deformation of the skin of the MLE system, the wing surface does not contribute to the structural resistance of the wing. What the authors called 'resistance' is the stiffness of the wing! They wrote that the main spar of the wing must be designed to withstand all the aerodynamic loads. This is wrong! The ribs work to transfer the aerodynamic load acting on the MLE to the main spar, so they must be able to bear the load. It is partially supported by the bending stiffness of the central region between the slits. But the MLE is not able to support torsion internal forces. Indeed the authors wrote that, to solve the problem of slits, the wing was covered with a sheet of heat-shrinkable plastic that don't provide any torsional stiffness.
They wrote that the sheet was placed in a way in which the motion of the LE was left free, in order to allow its morphing. So, the mechanism is design thinking to the kinematic requirement, but not to the structural requirement. This is an important issue because the MLE is able to move itself, but it is not able to bear the aerodynamic loads beacause the aerodynamic were not taken into accounto in the design process. For this reason there is not guarantee it works and it is able to maintain the maximum deflection shape during the wind tunnel test. Indeed, the morphed wing shape is not verified during the wind tunnel test and the morphed wing coordinates are computed by a FEM solver without considering the external aerodynamic loads.

- Third general comment: authors wrote that the proposed mechanism is a compliant mechanism. This could be true because it is based on the deformability of the ribs thank to the slits that must be properly sized. However, the authors present a 'design method' based on some equations that completely ignore the behaviour from the structural point of view. This aspect should be clarified.
For example, the authors wrote that the maximum displacement of the LE can be calculated by adding the contribution from each of the 6 slits, but the calculation of these contributions only depends on geometric considerations. I think it is not possible to estimate the deformation of a compliant system without any structural consideration and without considering the material properties. Why t, together with the material properties, are not apparently used in the y computation?

- Concerning the compliant mechanism, the authors wrote that the MLE has only one control arm because the wing ensures that there is no twist in the MLE and the three ribs move together with the same displacement'. But this is not true, because the ribs are flexible, they are not conventional ribs, so there are more than one degrees of freedom, due to the flexibility.

- FEA embedded into CATIA is a solid model, but the generation of FEM models inside CATIA is a completely automatic procedure and usually it requires a careful verification of the mesh. Please, add detailed informations about structural mesh, model and the structural analyses.
Concerning the structural analysis, it is impossible to understand what the authors mean when they write that the 'output was the constraint in the wood parts in MPa'. The results must be better explained and discussed.

- Two different validations are presented in the manuscript (the structural and the aerodynamic validation), but both contain many weaknesses.
STRUCTURAL VALIDATION
Authors wrote the value of this total displacement value can be numerically validated using the FEA of a morphed rib but no enought results about this validation are reported in the paper. In the author's opinion, the methodi to size the slits allows to design the MLE with a desired maximum displacement without needing a FEA. This is not true because they don't have a structural model in their procedure, and a high-fidelity FEM validation is always needed after the design. The authors don't need any optimization process, a simple structural model of the system is enough, and necessary!
Moreover, for all reasons decribed above both the mechanism design and the validation should be performed considering the aerodynamic loads.
AERODYNAMIC VALIDATION
First of all, the Aerodynamic simulation of the wing must be placed before the experimental tests in the manuscript.
The idea of using morphing to switch from an existing airfoil to an other existing airfoil, such as NACA airfoils, is interesting. However, some verification about the actual morphed shape that the system achieves is necessary.
The autors wrote that experimental distortions are larger than the distortion obtained using the FEA, so there is not a good correlation between the numerical models and experiment. The morphed shape should be estimated numerically considering the aerodynamic loads. Without this aeroelastic computation, the only way is a shape validation during the wind tunnel tests. One of these two approaches are necessary before performing the aerodynamic validation.

Author Response

We would like to thank very much to reviewer 2 for its comments. Please find below the comments of reviewer, and for each comment, our answers are given below.

The presented morphing mechanism is a variant of different shape adaptable leading edge concepts already proposed in literature. The most critical issue of this kind of mechanisms is the deformable skin. The authors didn't address this aspect neither they proposed a valid solution. Moreover the manuscript contains some interesting results, but together with many other things that are presented in a confusing and inconsistent way. For all these reasons a deep review of the whole manuscript is needed to publish it in a scientific journal.

The deep review should include the following issues providing a meticulous clarification.

Comment 2.1:

- First of all, using the name of the Price-Paidoussis Wind Tunnel in the title of a scientific is not appropriate. Moreover, the first piece of the introduction seems to be dedicated to sponsor Price-Paidoussis Wind Tunnel, while it should be used to present a method or a system, for example, in general.

Answer 2.1:

We removed the name of the Price-Païdoussis Wind Tunnel from the title of the article and replaced it by “a Subsonic Blow Down Wind Tunnel”.

The new title is “Design, Manufacturing and Testing of a New Concept of Morphing Leading Edge using a Subsonic Blow Down Wind Tunnel”.

The Price-Païdoussis Wind Tunnel is a type of subsonic blow-down wind tunnel and is not a commercial product. It was an important donation from McGill university Professors Michael P. Païdoussis and Stuart J. Price and we wanted to thank them for this donation by adding it in the title of the paper

The first part of the introduction gives a brief background on aeronautical research in Québec where CRIAQ (Consortium for Research and Innovation in Aerospace) is a major university-industry R&D grant initiative. It was not our intention to promote or sponsor a particular type of wind tunnel.

The first part of the introduction was moved to its end.

Comment 2.2:

- In the first sentence of the abstract authors wrote about 'this' morphing leading edge system before they introduce it. I suggest to replace 'this' whit 'a' generic system that is still to be presented. Moreover the following sentence should be removed 'the first part of this research concerns the design of a morphing trailing edge', as well as no results should be presented in the abstract.

Answer 2.2:

Thank you very much for the suggestions. The following sentences were rearranged:

L9: “This paper presents the design and wind tunnel tests results of a wing including a morphing leading edge for medium unmanned aerial vehicle with a wingspan of up to 5 m.”

L13: The sentence “the first part of this research concerns the design of a morphing trailing edge” was removed

L16: “modifying the stall angle of the wing up to 3°” was replaced by “modifying the stall angle of the wing”.

Comment 2.3:

- The paper is full of messages like 'Error! Reference source not found' that make manuscript reading very difficult.

Answer 2.3:

These errors were not in the original manuscript. We believe that they were introduced accidentally during the submission process. The references are now mentioned.

Comment 2.4:

- The first general aspect that must be clarified throughout the manuscript is that the proposed morphing concept can be only used as a wind tunnel model. It cannot be proposed as a general system suitable to be applied to the aircraft design. This because it is not able to bear the internal forces typical of an actual wing, such as the torsion (there is not a structural skin), and the external surface has many defects required by the system to work, but that would not be acceptable from the aerodynamic point of view on an actual aircraft.

Despite this, it seems that the authors propose this morphing mechanism as a general concept for all the aeronautical community. They wrote about the use of materials and actuators common in aircraft design and, again, about the prototype that was designed to validate the functionality of the deformation method, so the concept is general?
In my opinion the presented mechanism can be valid only to manufacture a wind tunnel model and to perform some aerodynamic test.

Answer 2.4:

Yes, this is not a general purpose morphing mechanism. It is designed to be integrated on UAV wings of a maximum of 5-meters wingspan. In order to clarify this point, we added the following sentences:

L9: The first sentence has been rewritten: “This paper presents the design and wind tunnel tests results of awing including a morphing leading edge for large unmanned aerial vehicle with their wingspan of maximum of 5 m.”

L27: We added “and integrating it on the wing of the UAS-S4 Ehécatl”

L85: We replaced “in aircraft design” by “in large model UAV design because of the fact that this system is mainly intended for UAV”

L556: We added “for a wingspan of 16.4 ft (5 m)”

Comment 2.5:

- Another general comment that must be clarified is related to the structural properties of the MLE. The authors wrote that to facilitate the deformation of the skin of the MLE system, the wing surface does not contribute to the structural resistance of the wing. What the authors called 'resistance' is the stiffness of the wing! They wrote that the main spar of the wing must be designed to withstand all the aerodynamic loads. This is wrong! The ribs work to transfer the aerodynamic load acting on the MLE to the main spar, so they must be able to bear the load. It is partially supported by the bending stiffness of the central region between the slits.

But the MLE is not able to support torsion internal forces. Indeed the authors wrote that, to solve the problem of slits, the wing was covered with a sheet of heat-shrinkable plastic that don't provide any torsional stiffness.

They wrote that the sheet was placed in a way in which the motion of the LE was left free, in order to allow its morphing. So, the mechanism is design thinking to the kinematic requirement, but not to the structural requirement.

This is an important issue because the MLE is able to move itself, but it is not able to bear the aerodynamic loads because the aerodynamic were not taken into account to in the design process.

For this reason there is not guarantee it works and it is able to maintain the maximum deflection shape during the wind tunnel test. Indeed, the morphed wing shape is not verified during the wind tunnel test and the morphed wing coordinates are computed by a FEM solver without considering the external aerodynamic loads.

Answer 2.5:

The loads are applied on the wing surface and they are transferred to the spar through the ribs. Therefore, all the loads are transferred to the fuselage through the spar. It means that the spar must sustain all the loads distributed along the span of the wing.
L92: “resistance” was a bad choice of word. The word “resistance” has been replaced by “strength”.

As seen in Figures 8 and 10, the slits cut the surface only at the leading edge of the wing; all the rest of the wing surface is covered with pieces of balsa sheet, which provide a very good torsional stiffness to the wing.

The analysis of external loads was already done during the MTE studies and it was found that for the size of our model at a speed of 20 m/s, the aerodynamics loads (less than 13.5 N for the entire wing) were neglected due to the loads provoked by to the deformation of the ribs.

The servomotor was strong enough to sustain the aerodynamic load applied on the leading edge. For this reason, we have not done the analysis as all the external loads for this model were considered. But it will be necessary to do this analysis when the MLE will be integrated on the UAS-S4 wing.

L154: An explanation was added on the way in which the wing can sustain the aerodynamic load applied on its leading edge: “the value of t was determined so that the servomotor can deform the rib until the slits close on themselves while maintaining the values of aerodynamic loads applied to the MLE”.

L156: We added an explanation on the way in which the aerodynamic loads were taken into account: “In order to take into account aerodynamic loads applied on the wing by using FEA in CATIA V5, a methodology was developed at LARCASE [17,18]”.
L158: Figure 12 representing the MLE with the aerodynamic loads around the wing was added: “Figure 12 shows the MLE with the aerodynamic load applied on its surface. The aerodynamic loads were obtained with XFLR5 software, and then, were imported into CATIA V5.”

The morphed wing shape was not verified in the wind tunnel because of the fact that it was found already that the obtained deformation was smaller in the model wing than the desired one. A rib LASER-cut alone deforms with the same amplitude as the FEA results. The wing model was a prototype that gave defects in its manufacture and caused mechanical interferences between the slits. These interferences reduced the deformation amplitudes of the MLE by 4 mm.

Comment 2.6:

- Third general comment: authors wrote that the proposed mechanism is a compliant mechanism. This could be true because it is based on the deformability of the ribs thank to the slits that must be properly sized. However, the authors present a 'design method' based on some equations that completely ignore the behaviour from the structural point of view. This aspect should be clarified.

For example, the authors wrote that the maximum displacement of the LE can be calculated by adding the contribution from each of the 6 slits, but the calculation of these contributions only depends on geometric considerations. I think it is not possible to estimate the deformation of a compliant system without any structural consideration and without considering the material properties. Why t, together with the material properties, are not apparently used in the y computation?

Answer 2.6:

The calculation considered that the material thickness at the slit t allowed the rib enough flexibility, so that the deformation was limited mechanically by the width of the slit. The value of t was used in the computation of y because of the fact that y = (lxL)/p (eq. (3)) with p = (e-t)/2 (eq. (1)).

L154: “the value of t was determined so that the servomotor can deform the rib until the slits close on themselves while maintaining the aerodynamic loads applied to the MLE”

Comment 2.7:

- Concerning the compliant mechanism, the authors wrote that the MLE has only one control arm because the wing ensures that there is no twist in the MLE and the three ribs move together with the same displacement'. But this is not true, because the ribs are flexible, they are not conventional ribs, so there are more than one degrees of freedom, due to the flexibility.

Answer 2.7:

The ribs were glued perpendicularly to the surface of the wing, so that they could not rotate relative to the surface, and remained parallel to each other. The rod allowed the force exerted by the servomotor to be splitted into forces on each of the three ribs.
The model control system was 9.5 in. wide, so that the possible twist of the leading edge was very small. But with a control surface at a bigger scale (around 20 in.), the use of two control arm could be necessary. Two arms could be connected to the same servomotor or to two servomotors acting in parallel in order to obtain a good rotation of the system.

Comment 2.8:

- FEA embedded into CATIA is a solid model, but the generation of FEM models inside CATIA is a completely automatic procedure and usually it requires a careful verification of the mesh. Please, add detailed information about structural mesh, model and the structural analyses.

Concerning the structural analysis, it is impossible to understand what the authors mean when they write that the 'output was the constraint in the wood parts in MPa'. The results must be better explained and discussed.

Answer 2.8:

Figure 13 representing the mesh was added.

The following explanation was added:

L164: “The mesh of the pieces that deform and of the other pieces (rib, arm, rod) are fine in order to obtain an accurate distribution of the constraints. The mesh of each part was generated with the tool “OCTREE Tetrahedron Mesh” with parabolic elements. The rib mesh had a “global size” of 0.15 in. and a “local size” around the slits of 0.02 in. The embedding of the assembly was placed on the two holes corresponding to the position of the main spar. The rib was connected to the control rod with a function “slider connection mesh”. The control rod had a mesh of a global size of 0.05 in. The control arm was connected to the control rod with the function “pressure fitting connection mesh”. It was modelled with a mesh of a “global size” of 0.1 in. and with a local mesh at both ends with a “local size” of 0.05 in. The control arm was connected with the servomotor with two function “slider connection mesh”. The servomotor had the largest global mesh of 0.25 in. because it did not deform. The servomotor was fixed to the rib with four function “contact connection mesh”. (Figure 13). In order to impose an angle of rotation for the servomotor head, a “rigid virtual part” was added on the control arm, connected with the servomotor head. A function “user-defined restraint” was placed on the “rigid virtual part” with a restrain on the rotation corresponding to the axis of the servomotor head. Finally, an “enforced displacement” was placed on the “user-defined restraint” with the rotation that wanted for the FEA. In Figure 13, the rotation angle was set to -5° to obtain a down motion of the leading edge. The mesh in the Figure 13 had a total of 61,393 elements.”

L186: “During the FEA, the input was the “angle of the servomotor head” in degrees, and the output was the “constraint” in the wood parts (ribs and control arm) in MPa (color from blue to red), as shown in Figure 14. The components that were the most stressed were the “control arm” and the “ribs”. The stresses on the rib were concentrated on the slits. As wood was used for the MLE design, the maximum stresses before the rib broke were around 70 MPa. The maximum stresses found in the control arm were around 30 MPa. Therefore, the control arm was able to send the rotation of the servomotor to the MLE. By concerning the slit, this FEA did not gave good results as the material in CATIA V5 was an isotropic one but the “wood” used for the ribs was “orthotropic”, but “orthotropic material 3D” was not available for computation. As the size of the slits were set following our design with the MTE, the flexibility of the ribs should be good. A static test of the flexibility of the ribs before wing manufacturing was done to ensure that the MLE would deform with the full amplitude desired.”

Comment 2.9:

- Two different validations are presented in the manuscript (the structural and the aerodynamic validation), but both contain many weaknesses.
STRUCTURAL VALIDATION

Authors wrote the value of this total displacement value can be numerically validated using the FEA of a morphed rib but no enough results about this validation are reported in the paper. In the author's opinion, the method to size the slits allows to design the MLE with a desired maximum displacement without needing a FEA. This is not true because they don't have a structural model in their procedure, and a high-fidelity FEM validation is always needed after the design. The authors don't need any optimization process, a simple structural model of the system is enough, and necessary!
Moreover, for all reasons described above both the mechanism design and the validation should be performed considering the aerodynamic loads.

AERODYNAMIC VALIDATION

First of all, the Aerodynamic simulation of the wing must be placed before the experimental tests in the manuscript.

The idea of using morphing to switch from an existing airfoil to an other existing airfoil, such as NACA airfoils, is interesting. However, some verification about the actual morphed shape that the system achieves is necessary.

The authors wrote that experimental distortions are larger than the distortion obtained using the FEA, so there is not a good correlation between the numerical models and experiment. The morphed shape should be estimated numerically considering the aerodynamic loads. Without this aeroelastic computation, the only way is a shape validation during the wind tunnel tests. One of these two approaches are necessary before performing the aerodynamic validation.

Answer 2.9:

By considering that t was sized in order to allow the full motion of the ribs, the FEA was not needed. Thus, the FEA was not needed to determine the maximum displacement of the leading edge tip, instead, the FEA was still used to determine the value of t for the design of the ribs.

The aerodynamic simulation was performed after the wind tunnel tests because of the fact that the MLE on the wing was designed using CATIA V5 and the mechanism was tested in the wind tunnel. An aerodynamic simulation of the model behavior in the wind tunnel was done. In this case, numerical results were validated using experimental results and this validation explained the reason why the simulation was performed after the experimental wind tunnel tests.

Another article will be submitted for its publication on a wing with the MTE and the MLE combined with the aim to morph the wing camber along its chord.

L385: “distortion” has been changed to “twist”.

Many thanks once again to reviewer 2 for his comments.

Round 2

Reviewer 2 Report

The revision the authors have made is clear and accurate, so my opinion is that tha manuscript can be now published in Biomimetics journal.